

# Tropospheric aerosols over the western North Atlantic Ocean during the winter and summer campaigns of ACTIVATE 2020: Life cycle, transport, and distribution

Hongyu Liu[1,2], Bo Zhang[1,2], Richard H. Moore[2], Luke D. Ziemba[2], Richard A. Ferrare[2], Hyundeok Choi[1,*], Armin Sorooshian[3], David Painemal[2,4], Hailong Wang[5], Michael A. Shook[2], Amy Jo Scarino[2,4], Johnathan W. Hair[2], Ewan C. Crosbie[2,4], Marta A. Fenn[2,4], Taylor J. Shingler[2], Chris A. Hostetler[2], Gao Chen[2], Mary M. Kleb[2], Gan Luo[6], Fangqun Yu[6], Jason L. Tackett[2], Mark A. Vaughan[2], Yongxiang Hu[2], Glenn S. Diskin[2], John B. Nowak[2], Joshua P. DiGangi[2], Yonghoon Choi[2,4], Christoph A. Keller[7,8], and Matthew S. Johnson[9]

[1]National Institute of Aerospace, Hampton, VA, USA
[2]NASA Langley Research Center, Hampton, VA, USA
[3]University of Arizona, Tucson, AZ, USA
[4]Analytical Mechanics Associates, Hampton, VA, USA
[5]Pacific Northwest National Laboratory, Richland, WA, USA
[6]State University of New York at Albany, Albany, NY, USA
[7]Morgan State University, Baltimore, MD, USA
[8]NASA Goddard Space Flight Center, Greenbelt, MD, USA
[9]NASA Ames Research Center, Moffett Field, CA, USA
[*]Now at SAIC / NOAA/NWS/NCEP/Environmental Modeling Center, College Park, MD, USA

Correspondence to: Hongyu Liu (hongyu.liu-1@nasa.gov); Bo Zhang (bo.zhang@nasa.gov)

**Abstract.** The Aerosol Cloud meTeorology Interactions oVer the western ATlantic Experiment (ACTIVATE) is a six-year (2019-2024) NASA Earth-Venture Suborbital-3 (EVS-3) mission to robustly characterize aerosol-cloud-meteorology interactions over the western North Atlantic Ocean (WNAO) during winter and summer seasons, with a focus on marine boundary layer clouds. This characterization requires understanding the aerosol life cycle (sources and sinks), composition, transport pathways, and distribution in the WNAO region. We use the GEOS-Chem chemical transport model driven by the MERRA-2 reanalysis to simulate tropospheric aerosols that are evaluated against in situ and remote sensing measurements from Falcon and King Air aircraft, respectively, as well as ground-based and satellite observations over the WNAO during the winter (Feb. 14 – Mar. 12) and summer (Aug. 13 – Sep. 30) field deployments of ACTIVATE 2020. Transport of pollution in the boundary layer behind cold fronts is a major mechanism for the North American continental outflow to the WNAO during Feb.-Mar. 2020. While large-scale frontal lifting is a dominant mechanism in winter, convective lifting significantly increases the vertical extent of major continental outflow aerosols in summer. Turbulent mixing is found to be the dominant process responsible for the vertical transport of sea salt within and ventilation out of the boundary layer in winter. The simulated boundary layer aerosol composition and optical depth (AOD) in the ACTIVATE flight domain are dominated by sea salt, followed by organic aerosol and sulfate. Compared to winter, boundary layer sea salt concentrations increased in summer over the WNAO, especially from the ACTIVATE flight areas to Bermuda, because of enhanced surface winds and emissions. Dust concentrations also significantly increased in summer because of long-range transport from North



Africa. Comparisons of model and aircraft submicron non-refractory aerosol species (measured by an HR-ToF-AMS)
vertical profiles show that intensive measurements of sulfate, nitrate, ammonium, and organic aerosols in the lower
troposphere over the WNAO in winter provide useful constraints on model aerosol wet removal by precipitation
scavenging. Comparisons of model aerosol extinction (at 550 nm) with the King Air High Spectral Resolution Lidar-
2 (HSRL-2) measurements (at 532 nm) and CALIOP/CALIPSO satellite retrievals (at 532 nm) indicate that the model
generally captures the continental outflow of aerosols, the land-ocean aerosol extinction gradient, and the mixing of
anthropogenic aerosols with sea salt. Large enhancements of aerosol extinction at ~1.5-6.0 km altitudes from long-
range transport of the western U.S. fire smoke were observed by HSRL-2 and CALIOP during Aug.-Sep. 2020. Model
simulations with biomass burning (BB) emissions injected up to the mid-troposphere (vs. within the BL) better
reproduce these remote-sensing observations, Falcon aircraft organic aerosol vertical profiles, as well as AERONET
AOD measurements over eastern U.S. coast and Tudor Hill, Bermuda. High aerosol (mostly coarse-mode sea salt)
extinction near the top (~1.5-2.0 km) of the marine BL along with high relative humidity and cloud extinction were
typically seen over the WNAO (< 35°N) in the CALIOP aerosol extinction profiles and GEOS-Chem simulations,
suggesting strong hygroscopic growth of sea salt particles and sea salt seeding of marine boundary layer clouds.
Contributions of different emission types (anthropogenic, BB, biogenic, marine, and dust) to the total AOD over the
WNAO in the model are also quantified. Future modeling efforts should focus on improving parameterizations for
aerosol wet scavenging and sea salt emissions, implementing realistic BB emission injection height, and applying
high-resolution models that better resolve vertical transport.

## 1.    Introduction

Aerosol particles scatter or absorb radiation in the atmosphere, directly affecting radiation budget and balance
and thus climate. Aerosols act as cloud condensation nuclei (CCN) or ice nuclei (IN), indirectly affecting radiation
via the formation of clouds and precipitation. They also affect tropospheric photolysis and photochemistry by
modifying solar radiation, and heterogeneous chemistry by providing surfaces for gas-particle interaction. While the
interaction of aerosols with clouds remains the largest uncertainty in the estimates of the Earth's changing energy
budget, a full understanding requires knowledge of aerosol transport, sources, sinks, composition, and distribution,
which still have large gaps (Boucher et al., 2013). Continental outflow regions represent a mixture of various aerosol
sources and types, and are impacted by large-scale and synoptic weather systems (Sorooshian et al., 2020), offering a
place for testing the current understanding of tropospheric aerosol processes. Among these regions, the western North
Atlantic Ocean (WNAO) presents "a complex atmospheric system with many unknowns" (Sorooshian et al. 2020;
Painemal et al. 2021). ACTIVATE is a six-year (2019-2024) NASA Earth Venture Suborbital-3 (EVS-3) mission to
investigate aerosol-cloud-meteorology interactions over the WNAO during winter and summer seasons, with a focus
on the marine boundary layer (MBL) clouds (Sorooshian et al., 2019; 2023). In this paper, we characterize the aerosol
life cycle, transport, composition, and distribution over the WNAO, defined as the oceanic domain bounded by 25–
50°N and 60–85°W, and North America's East Coast, during the winter (Feb.-Mar.) and summer (Aug.-Sep.)
campaigns of ACTIVATE 2020 using the GEOS-Chem chemical transport model.



Climatological circulation patterns largely determine the transport pathways and spatial distribution of trace
gases and aerosols over the WNAO (Sorooshian et al. 2020; Corral et al. 2021). The atmospheric circulation over the
North Atlantic Ocean basin is characterized by two semipermanent features: the Bermuda or Azores High (subtropical
anticyclone) and the Icelandic Low (subpolar low pressure) (Davis et al., 1997; Tucker and Barry, 1984). In summer,
the Bermuda High reaches its maximum spatial extent over the WNAO and extends westward, with southwesterly
winds over the western part of the domain north of 30°N and easterly trade winds in the subtropics (Painemal et al.,
2021). In winter, its expansion is limited by the development of the Icelandic Low north of 45°N. While the prevalent
westerly winds in winter/spring favor transport of pollution from North America to the Atlantic Ocean and toward
Europe, the trade winds south of the Bermuda High in summer facilitate transport of aerosols from the eastern Atlantic
and North Africa to the WNAO (e.g., Chen & Duce, 1983). The large-scale alternation of atmospheric mass between
the Bermuda High and the Icelandic Low leads to the so-called North Atlantic Oscillation (NAO; Lamb & Peppler,
1987), which plays a key role in modulating the transport of North American pollution to the northwestern Atlantic
Ocean and northern Europe (Christoudias et al., 2012). During the positive NAO phase (i.e., stronger Bermuda High
and Icelandic Low), anomalously stronger westerlies across the North Atlantic result in more North American
pollution over the WNAO and toward Europe, while during the negative NAO phase (i.e., weaker Bermuda High and
Icelandic Low), weaker westerlies lead to presence of less North American pollution in those regions.
On synoptic scales, the North American outflow of trace gases and aerosols to the WNAO is dominantly driven
by mid-latitude cyclones (Cooper et al., 2002; Li et al., 2005; Luan and Jaegle, 2013), which consist of four main
airstreams: warm conveyor belt (WCB), cold conveyor belt, dry airstream subsiding behind the cold front, and post
cold front boundary layer (BL) airstream (Carlson, 1998). Correspondingly, major transport mechanisms for North
American pollution outflow over the WNAO include horizontal advection within the BL behind the cold front, frontal
lifting by the WCB (ahead of the cold front), and convective lifting of continental pollution followed by westerly
transport in the free troposphere (Creilson et al., 2003; Li et al., 2002). For instance, Fast et al. (2016) identified key
processes responsible for the aerosol layers observed over Cape Cod, Massachusetts, and over the North Atlantic
Ocean (several hundred kilometers downwind) during the Two-Column Aerosol Project (TCAP) conducted during
July 2012. The aerosol layers observed in the free troposphere resulted from mean vertical motions associated with
synoptic–scale convergence ahead of a cold front, which lifted aerosols from the BL. Convection has been shown as
an effective mechanism for ventilating the U.S. continental BL, particularly in summer over the central and
southeastern U.S. (e.g., Li et al., 2005; Jacob et al., 1993; Dickerson et al., 1987). Recent aircraft observations from
the North Atlantic Aerosols and Marine Ecosystems Study (NAAMES) during 2015-2017 showed layers of sulfate,
black carbon, and organic aerosol enhancements in the free troposphere, suggesting long-range transport of continental
anthropogenic pollution and biomass burning (BB) emissions to the remote marine atmosphere (Croft et al., 2021).
Over the North Atlantic Ocean, North American pollution generally follows two transport pathways: one reaches
Europe in 4-5 days while the other is entrained in the Bermuda High anticyclone (Luan and Jaegle, 2013).
Tropospheric aerosols over the WNAO represent a mix of mainly anthropogenic, BB, biogenic, dust, and marine
emissions (see a comprehensive review by Sorooshian et al., 2020). The major aerosol types over the WNAO include
sulfate-nitrate-ammonium (SNA), black carbon (BC), organic aerosol (OA), dust, and sea salt. SNA aerosols are



mainly formed in the atmosphere through oxidation and neutralization of precursor gases sulfur dioxide ($SO_2$),
nitrogen oxides ($NO_x$), and ammonia ($NH_3$). They are water-soluble and subject to wet scavenging. Due to air pollution
regulatory policies in continental North America, anthropogenic $SO_2$ and $NO_x$ emissions have been significantly
reduced over the past couple of decades (Feng et al., 2020; Streets et al., 2006), resulting in a decreasing trend in fine
particulate matter mass concentrations and aerosol optical depth (AOD), as well as tremendous improvements in air
quality in the eastern U.S. and eastern Canada (van Donkelaar et al., 2019; Coen et al., 2020; Provençal et al., 2017;
Jongeward et al., 2016; Yang et al., 2018; Corral et al., 2021).

Sources of light-absorbing BC aerosols are both anthropogenic and natural (e.g., wildfires) in nature. OA is

either directly emitted (primary OA or POA) or formed in the atmosphere (secondary OA or SOA). SOA includes an
anthropogenic component from oxidation of aromatic hydrocarbons, and a biogenic component from oxidation of
biogenic volatile organic compounds such as terpenes. Many studies have characterized the impact of BB sources in
Alaska, western/Central Canada, western U.S. on the extended WNAO region, especially during the ICARTT and
TCAP field campaigns (Neuman et al., 2006; Berg et al., 2016). It has been shown that BB emission injection heights
involve large uncertainties (e.g., Pfister et al., 2006). Smoke plume heights derived from MISR/Terra observations
over North America ranged from a few hundred meters up to 5 km above the ground (Val Martin et al., 2010) and a
relatively high percentage of total BB emissions is injected above the BL in the North American boreal regions (Zhu
et al., 2018). Recently, Mardi et al. (2021) characterized BB aerosol events over the U.S. east coast and Bermuda over
the WNAO between 2005-2018 using ground-based and satellite observations in conjunction with MERRA-2
reanalysis data. More frequent BB events are found to occur in Jun.-Aug. over the northern part of the East Coast with
sources from western North America, while more events are identified in Mar.-May over the southeast U.S. and
Bermuda with sources from southern Mexico, Yucatan, Central America, and the southeast U.S. That study along
with others (Edwards et al., 2021) point to cloud-BB aerosol interactions over the East Coast and the WNAO. Long-
range transported North American wildfire aerosols, e.g., those from the Canadian wildfires in Aug. 2017 with the
extreme injection height of ~ 12 km, can be observed in the marine BL of eastern North Atlantic after descending in
the dry intrusion behind mid-latitude cyclones (Zheng et al., 2020). The August Complex "Gigafire" took place in
mid-August 2020 and the California Creek fire occurred in early September 2020, ranked among the top five in
California wildfire history (Zhuang et al., 2021). These fires are expected to have important impacts on trace gases
and aerosols, especially carbonaceous aerosols, over the east coast and the WNAO during the summer campaign of
ACTIVATE 2020.

Dust over the WNAO can be transported from North Africa, North America, and Asia. Dust over the northeast

U.S. is mainly transported in the lower and middle troposphere (2-6 km; Zhang et al., 2019) and long-range transport
of Asian dust in spring can reach the eastern US (Jaffe et al. 2003; DeBell et al., 2004). North African dust is
transported to the eastern US and the WNAO in summer (Jun.-Aug. maximum; Aldhaif et al., 2020) and the
trajectories are typically at ~ 1 km altitude (Savoie & Prospero, 1977; Perry et al., 1997). Contribution of North
American dust to the outflow to the WNAO is typically small (Corral et al., 2021). Modeling and observational studies
have found that an understanding of the dust loading and spatial (especially vertical) distribution over the WNAO is
still lacking (Colarco et al., 2003; Peyridieu et al., 2010; Generoso et al., 2008; Kim et al., 2014). Sea spray aerosols



are primarily generated by air bubbles bursting at the ocean surface resulting from wind stress and are composed of
inorganic sea salt and organic matter (de Leeuw et al., 2011; Quinn and Bates, 2014). Sea salt aerosols (SS) are a
major source of CCN, including giant CCN, over the WNAO (Gonzalez et al., 2022), and thus have indirect effects
on cloud, precipitation, and climate. As represented in MERRA-2, sea salt along with sulfate contribute most to total
AOD over the WNAO (Corral et al., 2021). MERRA-2 sea salt AOD over the WNAO is typically highest in winter
months and lowest in summer (Dadashazar et al., 2021; Aldhaif et al., 2021), consistent with sea salt mass
concentrations observed at IMPROVE sites along the U.S. East Coast (Corral et al., 2021). While sea salt is typically
the largest contributor to aerosol mass and extinction over the remote ocean, signatures of long-range transport of
anthropogenic, BB, and dust emissions are often present, as shown by Silva et al. (2020) in a model analysis of sun
photometer observations of AOD from two island sites over the North Atlantic.
The ACTIVATE mission deployed two aircraft (HU-25 Falcon and King Air) flying in coordination, with the
Falcon making in situ measurements in the lower troposphere and the King Air providing remote sensing
measurements of aerosols and clouds in the same vertical column from an altitude of 8-10 km while also launching
dropsondes. Flight hours totaled ~73 and ~60 for Falcon (~59 and ~67 for King Air) during the winter (Feb. 14 – Mar.
12, 2020) and summer (Aug. 13 – Sep. 30, 2020) deployments of ACTIVATE's first year of flights, respectively.
These intensive aircraft in situ and remote sensing observations of aerosols provide an opportunity to test the current
understanding of tropospheric aerosol sources and distribution as well as associated processes as represented by state-
of-the-art global models. In this paper, we evaluate the GEOS-Chem CTM driven by the MERRA-2 assimilated
meteorology (with marine POA emissions) against ACTIVATE aircraft measurements as well as ground and satellite
observations for the periods of the winter and summer campaigns of ACTIVATE 2020. We determine the sources,
transport, and distribution of tropospheric aerosols over the WNAO. The analysis also serves as a description of
aerosol conditions in the region during the two campaigns. We plan to address the following science questions: (1)
What are the major outflow pathways and transport mechanisms for the export of North American anthropogenic
pollution to the WNAO in winter and summer?; (2) Can a state-of-the-art chemical transport model reproduce the
distribution and variability of tropospheric aerosols over the WNAO as observed during ACTIVATE?; (3) What are
the sources of tropospheric aerosols as well as the relative contributions of terrestrial versus oceanic sources to the
aerosol mass, AOD, and their variability over the WNAO in winter and summer?; and (4) How is the summer
compared to the winter with respect to the sources, transport, and distribution of aerosols over the WNAO?
This paper is structured as follows. Section 2 introduces the GEOS-Chem model (with bulk aerosol) and model
simulations, and section 3 describes the observational data sets used in this study. Section 4 delineates the
meteorological setting and transport pathways for pollution over the WNAO. Section 5 presents the model simulated
aerosol composition and distribution over the WNAO (section 5.1) and model evaluations with aircraft in situ
measurements of CO, sulfate, nitrate, ammonium, and OA concentrations (section 5.2), AERONET AOD
measurements (section 5.3), and aerosol extinction profiles from aircraft HSRL-2 lidar and CALIOP/CALIPSO
satellite retrievals. Section 6 presents model source attributions of AODs over the WNAO during winter/summer
2020, followed by summary and conclusions in section 7.



## 2. Model Description


We use the GEOS-Chem global chemical transport model (www.geos-chem.org) version v11-01
(http://wiki.seas.harvard.edu/geos-chem/index.php/GEOS-Chem_v11-01) to simulate the sources, transport, and
distribution of tropospheric aerosols over the WNAO. The model is driven by the MERRA-2 assimilated meteorology
(at a horizontal resolution of 2°×2.5° with 72 levels) from the NASA Global Modeling Assimilation Office. It includes
a detailed description of stratospheric and tropospheric chemistry fully coupled through the Unified tropospheric-
stratospheric Chemistry eXtension (UCX; Eastham et al., 2014). Gas-phase tropospheric oxidant chemistry was
originally described by Bey et al. (2001) and its coupling with the SNA aerosol thermodynamics was developed by
Park et al. (2004). SNA thermodynamics are computed with the ISORROPIA thermodynamic module (Fontoukis and
Nenes, 2007). BC follows Wang et al. (2014), OA is after Pye et al. (2010) and Pye and Seinfeld (2010), and marine
primary OA (MPOA) uses the scheme of Gantt et al. (2015) based on monthly mean MODIS chlorophyll-a
concentrations. SOA follows the simplified Volatility Basis Set (VBS) scheme of Pye et al. (2010). Sea salt aerosol
emissions use the empirical source function of Jaegle et al. (2011) with a dependency on surface wind speed and sea
surface temperature. The model assumes two dry sea salt size bins, one for accumulation mode (radius 0.01-0.5 μm)
and the other for coarse mode (radius 0.5-8 μm). Dust emissions in GEOS-Chem were described by Fairlie et al.
(2007) and we use here the Dust Entrainment and Deposition (DEAD) scheme (Zender et al., 2003) with the size
distributions updated by Zhang et al. (2013). Aerosol optical depth is calculated for each aerosol type using local
relative humidity and prescribed optical properties (Martin et al., 2003; Drury et al., 2010; Ridley et al., 2012; Kim et
al., 2015). External mixing of aerosols is assumed. The input meteorological archives have 3-hour temporal resolution
for 3-D fields and 1-hour resolution for 2-D fields. The model time steps (10 min for transport and 20 min for
chemistry) are chosen to optimize both simulation accuracy and computational speed (Philip et al., 2016).
The model uses the TPCORE advection algorithm (Lin and Rood, 1996), computes convective transport from
the MERRA-2 convective mass fluxes (Wu et al., 2007), and uses the non-local scheme for BL mixing (Lin and
McElroy, 2010). The aerosol wet deposition scheme is described by Liu et al. (2001) and includes first-order rainout
and washout due to stratiform precipitation and scavenging in the convective updrafts. Scavenging of aerosol by snow
and cold/mixed precipitation is described by Wang et al. (2011, 2014). For stratiform precipitation scavenging, we
use the MERRA-2's spatiotemporally varying cloud condensed water content (CWC), following the revised scheme
of Luo et al. (2019, 2020), in the standard simulations in this study. For comparison, simulations prescribed with a
fixed CWC of $1.0\times10^{-3}$ kg m$^{-3}$ comparable to observed upper limits (Del Genio et al., 1996; Wang et al., 2011) are
also presented. On the other hand, MERRA-2 cloud cover and precipitation over the U.S. East Coast and WNAO are
biased low relative to satellite observations (Wu et al., 2022; Bosilovich et al., 2015, 2017) and thus introduce
uncertainty in the model scavenging processes. Aerosol dry deposition uses the resistance-in-series scheme of Wesely
(1989), with deposition to snow/ice surfaces from Fisher et al. (2011). Gravitational settling is as described by Fairlie
et al. (2007) for dust and Alexander et al. (2005) for coarse sea salt.
Global anthropogenic emissions are based on the EDGAR v4.2 emissions inventory for 2008. We use the 2011
EPA National Emissions Inventory (NEI11) emissions as implemented by Travis et al. (2016) over the U.S. with
national annual scaling factor updated for 2018 (Yu et al., 2018), CAC emissions inventory over Canada, BRAVO



emissions inventory over Mexico, EMEP emissions inventory over Europe, and the MIX emissions inventory (for
year 2010) over China (Li et al., 2014). Shipping emissions are from the EMEP inventory (Vestreng et al., 2007).
Aircraft emissions are from the AEIC inventory for 2005 (Stettler et al., 2011). The NEI11 emissions inventory
contains biofuel emissions. BB emissions are from the Quick Fire Emissions Dataset (QFED v2.5r1; Darmenov and
da Silva, 2015), which is based on the location and fire radiative power (FRP) obtained from the Moderate Resolution
Imaging Spectroradiometer (MODIS) Level 2 fire products and the MODIS geolocation products. QFED provides
daily mean emissions of trace gases and aerosols at 0.1°×0.1° horizontal resolution. BB emissions are injected within
the depth of the PBL in our standard simulations. In a separate set of simulations, they are also injected to the 0-5.5
km or 2-10 km altitude range to investigate the sensitivity of model results to BB emission injection heights. The
choice of these higher injection heights is based on the following previous studies. A substantial fraction of North
American fire emissions is injected to the free troposphere (e.g., val Martin et al., 2010). 35% of the QFED BB
emissions are also distributed between 3.5-5.5 km in the NASA GEOS-CF model (Keller et al., 2021; Fischer et al.,
2014). An explosive pyrocumulonimbus (pyroCb) cloud from Californian Creek fire on September 9, 2020 was
reported with the plume height peaking above 10 km (Carr et al., 2020). Biogenic emissions are calculated online
using the MEGAN model (Guenther et al., 2012). Marine DMS emissions are calculated as a product of the
climatological monthly mean surface seawater DMS concentration (Lana et al., 2011) and sea-to-air transfer velocity,
as implemented by Breider et al. (2017). Lightning $NO_x$ emissions are constrained by the Lightning Imaging Sensor
and the Optical Transient Detector (LIS/OTD) climatological observations of lightning flashes, as described by
Murray et al. (2012).
We perform model simulations for the periods of December 1, 2019 – March 31, 2020 and June 1 – September
30, 2020 with the first two months as the spin-up period. Model sensitivity experiments are also conducted to
investigate the impacts of using fixed CWC in model scavenging, sensitivity to BB emission injection height, and
impacts of various emission types (anthropogenic, BB, biogenic, marine, and dust). The impacts are quantified by the
difference in simulation results from the standard model and the sensitivity experiments. **Table 1** lists all model
experiments. Hourly, daily, and campaign-average output are saved for analysis.

**3.  Observational Data Sets**
**ACTIVATE aircraft data.** During ACTIVATE, the HU-25 Falcon aircraft made in situ measurements of
carbon monoxide (CO) mixing ratios and aerosol concentrations during each flight (duration of ~3.5 daytime hours).
CO measurements were made with a Picarro G2401 gas concentration analyzer (DiGangi et al., 2021). Submicron
non-refractory aerosol composition was measured by the High-Resolution Time-of-Flight Aerosol Mass Spectrometer
(HR-ToF-AMS; Aerodyne) (DeCarlo et al., 2008; Hilario et al., 2021), operated in 1 Hz Fast-MS mode and averaged
to 30 s time resolution. AMS data were collected downstream of an isokinetic double diffuser inlet (BMI, Inc.) and
also sampled downstream of a counterflow virtual impactor (CVI) inlet (BMI, Inc.) when in cloud (Dadashazar et al.,
2022; Shingler et al., 2012); only the former is used in this study. AMS measurements are reported at the standard
temperature (273.15 K) and pressure (1013.25 hPa). One-minute merged Falcon data are used in this study. We also
use the King Air's nadir-viewing High Spectral Resolution Lidar-2 (HSRL-2) retrievals of vertically resolved aerosol



extinction coefficient at 532 nm (Ferrare et al., 2023). The HSRL-2 instrument has been used in previous aircraft
missions and readers are referred to Burton et al. (2018) for further information about its operational details. The
FLEXPART model (Stohl et al., 1998) is used to identify the origin of air masses associated with high HSRL-2 aerosol
extinction during an event of the long-range transport of a western U.S. fire plume. ACTIVATE aircraft data and
FLEXPART model output are described in detail by Sorooshian et al. (2023) and available at:
https://doi.org/10.5067/SUBORBITAL/ACTIVATE/DATA001.
**CSN, IMPROVE, and NADP.** Surface aerosol composition ($SO_4^{2-}$, $NO_3^-$, $NH_4^+$) data for the eastern U.S. were
obtained from the Chemical Speciation Network (CSN) and the Interagency Monitoring of Protected Visual
Environments (IMPROVE) network (Solomon et al., 2014; Malm et al., 1994). IMROVE does not include $NH_4^+$
measurements. Daily observational data for 2020 were obtained from the Federal Land Manager Environmental
Database at the Cooperative Institute for Research in the Atmosphere (CIRA), Colorado State University
(https://views.cira.colostate.edu/fed; last access on 31 July 2022). The readers are referred to that website for methods
of chemical characterization, uncertainties, detection limits, and data downloading. Aerosol wet deposition
composition ($SO_4^{2-}$, $NO_3^-$, $NH_4^+$) data from the ensemble of eastern U.S. sites are from the National Trends Network
(NTN) of the U.S. National Atmospheric Deposition Program (NADP, https://nadp.slh.wisc.edu/). The NTN measures
total weekly wet deposition of $SO_4^{2-}$, $NH_4^+$, and $NO_3^-$ and we use monthly mean data downloaded from
http://views.cira.colostate.edu/fed/. Since the NTN wet deposition of $SO_4^{2-}$ includes contribution from $SO_4^{2-}$ resulting
from oxidation of $SO_2$ in rainwater, for comparison with NTN measurements we calculate the model sulfur wet
deposition fluxes as the sum of the $SO_4^{2-}$ wet deposition flux and 150% (based on the ratio of the molar masses of $SO_2$
and $SO_4^{2-}$) of the model estimated $SO_2$ wet deposition (Appel et al., 2011). Similarly, the model estimates of $NO_3^-$
($NH_4^+$) wet deposition include 98.4% (106%) of the model estimates of $HNO_3$ ($NH_3$) wet deposition. Gartman (2017)
described the analysis and quality assurance of the NADP data. The IMPROVE and NADP data sets were previously
used by Corral et al. (2020, 2021) to analyze aerosols and wet deposition chemistry over the U.S. East Coast.
**AERONET and MODIS.** We use AOD measurements from the Aerosol Robotic NETwork (AERONET,
http://aeronet.gsfc.nasa.gov; Holben et al., 1998), a ground-based aerosol remote sensing network. Level 2.0 daily
data used are based on the Version 3 algorithm and are cloud screened and quality assured (Giles et al., 2019; Smirnov
et al., 2000). AERONET AOD data were obtained for three sites during Feb.-Mar. and Aug.-Sep. 2020: NASA LaRC
(37.10°N, 76.38°W), NASA GSFC (38.99°N, 76.84°W), Tudor Hill, Bermuda (32.26°N, 64.88°W). For comparison
with GEOS-Chem AODs at 550 nm, AERONET AOD values at 440 nm were converted to 550 nm using the
AERONET 440-675 nm Angstrom exponent. Estimated uncertainties in AERONET AODs are on the order of ~0.01-
0.02 (Eck et al., 1999; Dubovik et al., 2000). Satellite retrieved products of AODs are available from MODIS on Aqua.
MODIS has a ~2330 km swath and provides nearly global coverage daily. We use the MODIS/Aqua Level 3 daily
(MYD08_D3) AOD product at 550 nm and 1°×1° horizontal resolution (Collection 6.1; scientific data set or SDS
named "Aerosol_Optical_Depth_Land_Ocean_Mean") from the Dark Target (DT) retrieval algorithm (Sayer et al.,
2014; Levy et al., 2013; Hubanks et al., 2019). This SDS contains only AOD values for any DT-ocean retrieval having
the Quality Assurance Confidence (QAC) >=1 and any DT-land retrievals having QAC=2 (good confidence) or 3



(high confidence) (Levy et al., 2013; R.C. Levy, personal communication, 2019). Collection 6.1 data are available
from https://ladsweb.modaps.eosdis.nasa.gov/search/.
**CALIPSO.** The Cloud-Aerosol Lidar with Orthogonal Polarization (CALIOP), on board the Cloud-Aerosol
Lidar and Infrared Pathfinder Satellite Observations (CALIPSO) platform, has been providing aerosol vertical profile
measurements of the Earth's atmosphere on a global scale since June 2006 (Winker et al., 2010). We use the version
4.51 CALIOP Level 2 Aerosol Profile products with a vertical resolution of 60 m and horizontal resolution of 5 km
over an altitude range of 30 km to -0.5 km, and only quality screened extinction samples are used in the analysis.
Specifically, aerosol layers with Cloud Aerosol Discrimination (CAD) scores less than -100 or greater than -20 are
rejected to avoid low-confidence aerosol classifications (Liu et al., 2019). Also, aerosol layers with the extinction
Quality Control (QC) flag not equal to 0, 1, 16, and 18 are rejected to remove low-confidence extinction retrievals,
while aerosol extinction samples with the extinction uncertainty equal to 99.99 km$^{-1}$ as well as those at lower altitudes
below these samples are rejected to remove unreliable extinction values (Yu et al., 2010; Winker et al., 2013; Kim et
al., 2018; Tackett et al., 2018).
In addition, we apply the same data averaging approach that was used to generate the CALIPSO version 4 Level
3 aerosol products (Tackett et al., 2018). The following quality screening and data selection techniques are applied.
1) When creating the averaged profiles used in this work, CALIOP Level 2 range bins in "clear air" regions
where no aerosol is detected are typically assigned aerosol extinction coefficients of 0 km$^{-1}$. An exception to this
convention is exercised for regions of "clear air" lying below aerosol layers with base heights lower than 250 m.
Because we assume the atmosphere is well mixed below 250 m, excluding these bins avoids introducing low biases
near the Earth's surface.
2) Aerosol layers detected at 80 km horizontal resolution that are not vertically or horizontally adjacent to another
aerosol layer detected at finer spatial resolutions (i.e., 5 km or 20 km) are assumed to be noise-induced false positive
detections, and hence are discarded.
3) Layers classified as aerosols that are detected at 80 km horizontal resolution with base altitudes above 4 km
and are in contact with ice clouds are assumed to be misclassified cirrus "fringes" (Liu et al., 2019), and are also
discarded.
4) Aerosol samples in the surface-attached opaque layer (beneath the maximum surface elevation), for which
the extinction is greater than 2 km$^{-1}$ and at least 10 times higher than the extinction of the layer above, are rejected to
avoid surface contamination.
5) In any 5-km profile in the CALIOP Level 2 data product, range bins identified as aerosol that have embedded
cloud fractions greater than 97% are rejected, as are all samples at lower altitudes in the profile. Rejecting these
samples reduces the likelihood of cloud contamination.
The details of the quality screening to generate the Level 3 aerosol profile product are described in Winker et al.
(2013), Tackett et al. (2018), and the CALIPSO Data Users' Guide (https://www-
calipso.larc.nasa.gov/resources/calipso_users_guide/). CALIOP aerosol extinction coefficients at 532 nm were
horizontally and vertically regridded onto the GEOS-Chem grids by averaging all quality screened extinction values
within each grid box. CALIOP data are available at https://subset.larc.nasa.gov/calipso.




## 4.  Meteorological Settings and Transport Pathways

In this section we describe the mean states of meteorological settings, BL outflow, and vertical transport of pollution during Feb.-Mar. and Aug.-Sep. ACTIVATE 2020, as represented by the MERRA-2 reanalysis and GEOS-Chem model simulation. They will facilitate the interpretation of model results as well as comparisons with observational data in section 5. **Fig. 1** shows the tracks of 22 HU-25 (Falcon) flights during Feb.-Mar. ACTIVATE 2020 and 18 flights during Aug.-Sep. ACTIVATE 2020, with flights typically transiting via waypoints of ZIBUT (36.93°N, 72.67°W) and OXANA (34.36°N, 73.75°W) to avoid military restricted air space. The aircraft sampling domain is divided at 36°N into two box regions: the north ("N"; 36-39°N, 69-75°W) and the south ("S"; 32.5-36°N, 71-75.5°W) for this analysis.

**Meteorological settings.** The winter and summer campaign periods feature contrasting atmospheric circulation patterns and meteorological conditions. **Fig. 2a** and **Fig.2b** show MERRA-2 near surface air temperature, surface level pressure, relative humidity, vertical pressure velocity ($\omega$), total and convective precipitation, and PBL heights (based on the total eddy diffusion coefficient of heat with a threshold value of 2 m$^2$ s$^{-1}$; McGrath-Spangler and Molod, 2014) over the WNAO averaged over Feb.14-Mar.12 and Aug.13-Sep.30, respectively. Also shown in the figures are the model simulated CO concentrations and aerosol extinction coefficients (at 550 nm) at 945 hPa. In winter, the lower troposphere of the WNAO region was dominated by westerly wind and air masses from continental North America in the north and anticyclonic winds in the southeast. The latter was associated with the Bermuda High located to the east of the domain. A strong NW-SE horizontal gradient in surface temperature extended from the SE U.S. coast towards NE/E, consistent with frequent passages of cold fronts driven by the Northern Hemisphere mid-latitude cyclones and the warm Gulf Stream sea surface temperatures in Feb.-Mar (e.g., Seethala et al., 2021). The ACTIVATE flights in the two-box region sampled this continental outflow extensively. Flights to the north often occurred during post-frontal conditions, reflecting one of the mission objectives to sample and study MBL clouds, especially during cold air outbreaks in winter that have been the subject of recent work (Tornow et al., 2022; Seethala et al., 2021; Corral et al., 2022; Li et al., 2022; Chen et al., 2022). In addition to lower temperature, the post-frontal air was characterized by subsiding motion with lower relative humidity and precipitation (dry air) as well as lower BL height. In particular, the S box region experienced stronger uplifting and precipitation (predominantly stratiform) ahead of the cold fronts. It suggests stronger aerosol scavenging associated with the southern cluster of flights. Model BL CO concentrations showed large land-to-sea gradients over the WNAO, resulting from the fact that North American pollution outflow was much stronger in the N box region (westerly winds) than in the S box region, which was often intruded by low-latitude marine air (southwesterly winds), during Feb.-Mar. 2020. The more inhomogeneous distribution of aerosol extinction coefficients (compared to CO) generally reflects the shorter aerosol lifetime, as well as more complex processes and their interactions involved in speciated aerosol emissions, transport, heterogeneous chemistry, photochemistry, and wet scavenging.

In summer, while midlatitude cyclones and westerlies move northward, the Bermuda High strengthened and extended westward with southwesterly offshore along the U.S. SE coast (~32-36°N) and easterly trade winds in the subtropics (<30°N; **Fig. 2b**). Compared to the wintertime, the horizontal gradients in surface temperature, RH, vertical



pressure velocity, precipitation, and PBL height were much weaker in the N and S box regions. However, convective precipitation was much stronger in summer and accounted for ~50-80% of the total precipitation in the flight domain. Despite a large land-to-sea gradient, model simulated BL CO concentrations were much lower due to CO oxidation by higher OH in summer. In contrast to the wintertime, BL aerosol extinction over the WNAO during summer exhibited large enhancements compared to those over land, suggesting major aerosol sources of marine origin.

**Boundary-Layer outflow of pollution.** Transport in the BL behind cold fronts is a major mechanism for the North American pollution outflow to the WNAO. It exerts large impacts on the BL trace gases and aerosol composition as well as their spatiotemporal evolution in the ACTIVATE flight domain. **Fig. 3** shows Hovmöller diagrams of model daily mean air temperature, CO, and sea salt concentrations at 950 hPa along 72.5°W (near the longitude of waypoint ZIBUT) over the WNAO during Feb. 14 - Mar. 13 and Aug. 13 – Sep. 30. During Feb.-Mar. 2020 (left column, **Fig. 3**), there were about four major periods with cold fronts passing through the study area, as indicated by the wavy pattern of near-surface temperature day-to-day variations. During frontal passages, CO pollution was swept in the BL behind the cold fronts southeastward to the ACTIVATE flight latitudes (32.4-39.8°N). On Feb. 15-16, Feb. 28, Mar. 1, and Mar. 7, the BL outflow of CO reached as far as south of 32°N. On the other hand, advection of warm marine air from low-latitudes resulted in low CO concentrations across the flight domain. Enhanced sea salt aerosols were often associated with strong wind and warm air from the south (left bottom panel, **Fig. 3**). A remarkable sea salt event occurred because of strong surface wind speed on March 7 during a cold air outbreak. During Aug.-Sep. 2020 (right column, **Fig. 3**), cold air intrusion deep into the flight domain did not take place until late September since midlatitude cyclones were shifted northward in summer. High CO events within the ACTIVATE flight latitude ranges appeared associated with transport of anthropogenic or wildfire emissions that occurred between 35-45°N. Two strong events of high sea salt concentrations occurred on September 10-12 and 19-22. The former was due to high surface winds associated with a westward moving low-pressure system. The latter resulted from the strong winds during a cold air outbreak that lasted for several days when cold air swept along a NE-to-SW corridor off the east coast over the whole flight domain. It lasted until a new cold front moved into the domain on September 23, resulting in enhanced CO during September 23-25.

**Vertical Transport.** Major pathways for vertical transport of trace gases and aerosols over the North American continental outflow region include uplifting ahead of cold fronts, convective transport, and BL turbulent mixing. **Fig. 4** shows model resolved large-scale vertical fluxes, convective fluxes, and PBL turbulent mixing fluxes of CO (**Fig. 4a&4b**) and sea salt (**Fig. 4c&4d**) at 1 km and 2 km, respectively, averaged over the periods of Feb. 14 - Mar. 12 and Aug. 13 – Sep. 30, respectively, 2020. During Feb.-Mar. 2020, large-scale vertical transport ahead of cold fronts apparently played a dominant role in lifting CO out of the BL over the U.S. east coast and two flight box regions (**Fig. 4a**), followed by rapid eastward transport in the free troposphere. Convection was also important in this role, especially in the S box region. It became even more important in summer during Aug.-Sep. 2020 when convective fluxes of CO were comparable to large-scale vertical fluxes (**Fig. 4b**). For sea salt, by contrast, BL turbulent mixing was found to be the dominant process responsible for upward transport of sea salt within and ventilation out of the BL over the flight domain in winter (**Fig. 4c**), while both convection and turbulent mixing were important in uplifting sea salt to the free troposphere in summer (**Fig. 4d**).



## 5. Simulated aerosols over the WNAO and model evaluations

### 5.1 Simulated aerosol composition, distribution, and loading over the WNAO

In this section, we describe the horizontal, vertical distributions, and mass loadings of aerosol species over the WNAO during Feb.-Mar. and Aug.-Sep. 2020, as simulated by GEOS-Chem. **Fig. 5** shows the 929 hPa maps of model simulated mass concentrations of sulfate-nitrate-ammonium (SNA), black carbon (BC), primary organic aerosol (POA), secondary organic aerosol (SOA), sea salt, and dust averaged over the period of Feb. 14 - Mar. 12, 2020. Also shown are longitude-altitude cross-sections of each aerosol species averaged over 33-39°N. The distribution patterns of SNA, BC, POA, and SOA resemble those of CO (**Fig. 2a**), reflecting the frequent North American continental outflow of traces gases and aerosols in the BL behind cold fronts to the WNAO. Among aerosol species of North American origin, POA shows the highest mass in the study domain while POA and nitrate exhibit strong gradients with mass concentrations sharply decreasing eastward. As will be discussed in section 5.2, POA in the study domain are attributed to North American anthropogenic, biofuel, and southeast U.S. BB emissions. Sulfate mass concentration shows a weaker gradient because DMS from ocean is also a source of $SO_2$ and sulfate. Sea salt has the largest aerosol mass over the WNAO, with maximum in the easterly trade wind region and to the east of the north flight domain. Abundances of BC, SOA, and dust across the flight domain are relatively small. Dust to the southeast of Bermuda is a result of long-range transport from Africa. As shown in the vertical cross-section plots, aerosol masses are mostly restricted in the lower troposphere (< 2-3 km).

**Fig. 6** shows the same plots as in **Fig. 5**, but for Aug. 13 – Sep. 30, 2020. Sulfate concentrations in the BL increased significantly relative to winter because of stronger production of sulfate from oxidation of $SO_2$ in summer. The stronger west-to-east gradient in sulfate concentrations was partly due to the lack of cold front passages as midlatitude cyclones shift to higher latitudes in summer. Nitrate concentrations in the BL decreased substantially due to higher temperatures that limit particle-phase ammonium nitrate (a major chemical form of nitrate associations) as well as the competition for ammonium by more sulfate. The simulated high nitrate concentrations in the upper troposphere between 33-39°N are presumably from lightning $NO_x$ emissions. The spatial pattern of BC concentrations over land is consistent with a major source from the western U.S. BB in summer (versus primarily from anthropogenic and biofuel emissions in winter). However, BC remains a small contribution to the outflow of aerosols. POA concentrations are high in the BL albeit lower than in winter, with a peak in the lower free troposphere (at ~2 km altitude) over the WNAO. Its primary sources are North American BB and anthropogenic emissions. Much higher SOA concentrations reflect the larger production from strong oxidation of biogenic VOCs from the southeast US in summer. The vertical extent of the major North American continental outflow aerosols was significantly higher than that in winter, reflecting the impact of summertime convective lifting. BL sea salt concentrations increased in summer over the WNAO, especially from the ACTIVATE flight areas to Bermuda, because of stronger winds (section 4). Dust amounts also significantly increased because of long-range transport of dust emissions from North Africa. It is noted that African dust can be transported to the Gulf of Mexico and then northward to the eastern U.S., as shown by enhanced dust concentrations in the BL between 80-85°W (dust vertical cross-section, **Fig. 6**).





As a summary of model simulated aerosols, **Fig. 7** shows model simulated mean concentrations (µg m$^{-3}$ STP) or
AODs of each aerosol species (bar graph) and their fractions of the total aerosol mass or AODs (pie charts) for the
flight areas ("N" and "S" in **Fig. 1**) over the periods of Feb. 14 – Mar. 12 and Aug. 13 – Sep. 30 ACTIVATE 2020,
respectively. Sea salt is a dominant fraction of total aerosol mass in the BL in both winter (53%) and summer (72%),
followed by organics with 24% in winter and 13% in summer. Higher sea salt mass concentration is also the main
reason for higher total aerosol mass in summer. SNA is about 20% of the total aerosol mass in winter and 11% in
summer. Dust accounts for 1% in winter and 3% in summer, with the latter season's increase reflecting more efficient
transport from the eastern Atlantic and North Africa by the subtropical trade winds. Sea salt, SNA, and OA are the
main contributors to the mean total AOD in the flight areas, with the following percentage contributions: SS: 41%
(winter), 62% (summer); OA: 26% (winter), 16% (summer); and SNA: 31% (winter), 20% (summer). The much larger
fraction of AOD (versus aerosol mass) from SNA is ascribed to the strong hygroscopic growth of those fine aerosols,
similar to OA.

**5.2 Model evaluation with aircraft in situ measurements and source analysis**
In this section we evaluate model simulations of CO and aerosols with ACTIVATE aircraft in situ measurements.
A series of GEOS-Chem experiments with different configurations were performed to investigate the impacts of key
aerosol-related processes in the model, including emission sources (anthropogenic and biofuel, BB, and marine), BB
emission injection height, and cloud water content (fixed value vs. MERRA-2, **Table 1**). **Fig. 8** and **Fig. 9** compare
model simulated vertical profiles of CO, SNA, and OA mixing ratios with Falcon aircraft Picarro and AMS
measurements during Feb.-Mar. and Aug.-Sep. 2020, respectively. It is noted that the observed profiles above 3 km
are probably biased because Falcon aircraft flew below 3 km most of the time. Also shown in the figures are model
results from simulations (**Table 1**) with (1) a fixed value for cloud water content used in aerosol scavenging
("fixedCWC"), (2) BB emissions injected to the 0-5.5 km altitudes (for Aug.-Sep. 2020 only), (3) fossil fuel and
biofuel emissions turned off, (4) BB emissions turned off, or (5) marine emissions turned off, respectively. Values
(500 m-binned) are medians over all flights.
In winter, the aircraft observations showed a decreasing trend of CO, SNA, and OA aerosol concentrations with
altitude, with substantially higher concentrations in the BL as part of the strong North American continental outflow
of pollution and a sharp vertical gradient at ~1.0 km above the sea surface. The sharper gradient in aerosols than in
CO concentrations suggests that aerosols were scavenged during uplifting processes. The simulated profile of CO
concentrations is in good agreement with the observations. The model captures the observed sulfate concentrations in
the BL but underestimates them in the free troposphere, likely due to uncertainty in the model aerosol scavenging
scheme, as discussed below. Simulated nitrate concentrations in the BL are too high compared to observations. The
model reasonably captures the vertical distribution of ammonium concentrations, but the vertical gradient in the lower
troposphere is too strong. The simulated OA concentrations are biased high in the BL and biased low above ~4 km.
The model attributes CO and SNA aerosols mainly to anthropogenic emissions, as indicated by the large reduction in
their concentrations compared to the standard simulation (red lines, **Fig. 8**) when anthropogenic emissions are turned
off in the model (green lines, **Fig. 8**). The model suggests that while both anthropogenic and BB emissions are major



contributions to OA in the BL, BB is likely responsible for the OA enhancement at ~4.0 km, for which the model
predicted it at a lower altitude (~3.5 km). Injecting BB emissions to the altitude range of 0-5.5 km results in OA peak
concentrations (at ~3.5 km and ~5.0 km) much higher than observed, suggesting the 0-5.5 km injection height is too
high in winter (dark yellow line, **Fig. 8**). The effect of marine emissions on SNA and OA is small in the model.
Using the MERRA-2 interactive CWC (versus a fixed value) for the aerosol wet scavenging scheme in the model
has a large impact on simulated aerosol profiles over the flight domain. As shown in **Fig. 8** (blue lines), using the
MERRA-2 CWC enhances aerosol scavenging and results in lower aerosol concentrations in the troposphere. This is
mainly because generally the MERRA-2 CWCs for precipitating stratiform clouds are significantly smaller than the
assumed fixed value ($1.0\times10^{-3}$ kg m$^{-3}$, **Table 1**), leading to a faster conversion from cloud water to the same amount
of precipitation and thus stronger scavenging. SNA aerosols are affected more compared to OA because they are more
soluble. The reduction in sulfate concentrations is substantial, especially in the middle and upper troposphere. The
model overestimate of sulfate in the MBL compared to aircraft AMS measurements is corrected by this update in wet
scavenging, but simulated concentrations above ~4 km are far too low, suggesting ice scavenging is too fast. This low
bias seems not as obvious for nitrate or ammonium, presumably due to the SNA thermodynamic equilibrium where
lower sulfate favors more nitrate associated with ammonium in the aerosol phase.
In summer, aircraft measured CO concentrations showed a relatively weak vertical gradient with much lower
concentrations in the BL compared to winter and enhancement layers in the middle and upper troposphere (~3-4 km
and ~5.5-6.5 km; **Fig. 9**). The former reflects the shorter CO lifetime in summer and the latter results from the long-
range transport of North American continental pollution as well as the western U.S. fire emissions. Observed sulfate
concentrations exhibited a strong vertical gradient with much higher levels in the BL compared to winter along with
enhancements at 3-4 km. The generally higher sulfate concentrations throughout the troposphere in summer reflect
stronger oxidation of $SO_2$ in both the gas phase and in-cloud (Dadashazar et al., 2022; Tai et al., 2010). The observed
nitrate concentrations in the BL have a median value close to that in winter but show much smaller variability. BL
ammonium and OA concentrations are substantially higher than in winter. All of nitrate, ammonium, and OA
observations show large enhancements between ~3.5 km and ~5 km.
The model reasonably reproduces the observed CO concentrations in the BL but fails to capture the observations
in much of the free troposphere, especially the magnitude of CO enhancements around ~4 km and ~6 km (red and
black lines, CO panel of **Fig. 9**). The reason for the latter is attributed to the model's inability to retain the western
U.S. BB CO plumes when reaching the eastern U.S. and the ACTIVATE flight areas. Nevertheless, clear reductions
of CO in the model simulations without BB or anthropogenic sources suggest both are important sources to CO in
summer over the WNAO.
For SNA and OA aerosols, the standard model simulation tends to underestimate the observed concentrations in
the BL in contrast to the wintertime (red and black lines, aerosol panels of **Fig. 9**). The model also underestimates
observations in most of the free troposphere, except for nitrate. The observed nitrate and OA layer enhancements
between 3-5 km are reasonably simulated, with the OA peak at slightly lower altitude (~3.5 km) and of much lower
concentration in the model. On the other hand, the model barely captures the sulfate and ammonium enhancements
observed in this layer. All these underestimated enhancements are largely improved or corrected by extending the BB





emission injection height from within the BL to 0-5.5 km (dark yellow lines, **Fig. 9**), suggesting that releasing the BB
emissions within the BL significantly limits the long-range transport of fire emissions in the free troposphere. The
model simulates too high nitrate concentrations in the mid-upper troposphere (>5 km). This is presumably due to the
large reduction in sulfate resulting from the use of MERRA-2 CWC in the aerosol scavenging scheme (red and blue
lines, sulfate panel of **Fig. 9**). Less sulfate shifts the SNA balance to favor nitrate in the model as more nitrate is
retained in the aerosol phase to neutralize ammonium in the SNA system. The model sensitivity experiments suggest
that SNA aerosols are predominantly from continental anthropogenic emissions in summer like winter. However, in
summer the BB emissions are also important sources of SNA between ~3-6 km. BB is the dominant source of OA
(mostly primary) in the free troposphere, whereas BB, anthropogenic, and marine emissions all contribute to OA in
the marine BL (with SOA accounting for less than one-third below 1 km). The large model underestimate of OA in
the BL is likely due to weak entrainment from the free troposphere, low production of SOA, and/or strong scavenging
associated with convective precipitation. The effect of marine emissions on sulfate appears more significant in the BL
and lower free troposphere compared to winter, reflecting the stronger oxidation of marine DMS and convective lifting
of DMS or its oxidation products ($SO_2$, sulfate) in summer.

**5.3 Model evaluation with AERONET AOD measurements**


Comparisons of model results with surface aerosol concentration observations from the IMPROVE and CSN
networks, aerosol deposition flux measurements from the NTN network of NADP, and satellite AOD measurements
from MODIS/Aqua are included in the **Supplement (section S1)**. In this section we evaluate model simulated AOD
with ground-based measurements from AERONET, with a focus on the eastern U.S. coastal region.
Continuous measurements of AOD from AERONET are used to evaluate the model performance in reproducing
the observed AOD magnitude and temporal variability as well as long-range transport of aerosol plumes. **Fig. 10**
shows model simulated daily AOD (at 550 nm) versus daily AOD measurements from two AERONET sites (NASA
LaRC, NASA GSFC) in the eastern U.S. and one site (Tudor Hill) located at Bermuda over the Atlantic Ocean during
Feb.-Mar. and Aug.-Sep., respectively, 2020. Model results are from the simulations (**Table 1**) "standard",
"fixedCWC" (for winter only), "BB0-5.5km" (for summer only), "noanthbf", "nobb", and "nomari", respectively.
Also shown in the figure are scatterplots of model AODs from the "standard" simulation versus AERONET AODs at
the three sites for winter and summer, respectively. In winter, AERONET AODs at LaRC and GSFC show large day-
to-day variability with an increasing trend towards early spring. Those at Tudor Hill exhibit smaller day-to-day
variability but much larger variability on a weekly timescale. The standard model simulation (black lines) reproduces
the observations at the sites reasonably well. Using a fixed value for CWC in the wet scavenging scheme (green lines)
significantly degrades the model performance. Comparisons between the standard and sensitivity simulations suggest
that AODs at LaRC and GSFC are mainly attributed to anthropogenic emissions. Contributions from BB emissions
are significant and may become comparable to anthropogenic contributions in some days. AODs at Tudor Hill are
mostly ascribed to marine (sea salt) emissions but are also affected by anthropogenic emissions, presumably from
North America. In summer, AERONET AODs at LaRC and GSFC indicate even larger day-to-day variations with
larger maximum AODs (~0.3-0.4) compared to winter (~0.2-0.3). The minimum AODs tend to decrease with time.



At Tudor Hill, AERONET AODs indicate high values (~0.2-0.3) in early and late Aug. (data not available for most
of Sep. at the time of this study). AERONET AODs at all three sites clearly identify two extremely large AOD events
close to Aug. 26 and Sep. 23; the western U.S. fire smoke associated with these events were also observed by aircraft
during ACTIVATE (section 5.4; Mardi et al., 2021; Corral et al., 2022). The standard model (black lines) reproduces
the decreasing trend of minimum AODs at LaRC and GSFC, but largely underestimates the magnitude of AODs at
LaRC with better performance at GSFC. The simulation "BB0-5.5km" (orange lines) where BB emissions are injected
between 0-5.5 km significantly improves the model's capability of capturing the large AOD events, especially at
GSFC. Comparison between the standard and sensitivity model simulations suggest that while AODs are often
attributed to anthropogenic (and BB to a lesser extent) sources, BB emissions are mainly responsible for the large
AOD events on or around Aug. 26 and Sep. 23. For the Aug. 26 event, the large reduction in model AODs when BB
emissions are turned off indicates that the transport of the western U.S. fire smoke plumes to Tudor Hill is successfully
captured by the model. The model attributes the observed large AODs in early Aug. to long-range transport of dust
from northern Africa. However, most of the time AODs at Tudor Hill are mainly due to marine (sea salt) emissions.
The above analysis suggests that GEOS-Chem simulated AODs and their variability in the WNAO region are
reasonable.

**5.4 Model evaluation with aerosol extinction profiles from aircraft HSRL-2 lidar and CALIOP/CALIPSO satellite retrievals**

The NASA LaRC HSRL-2 lidar instrument on the King Air aircraft measured aerosol extinction profiles in the
same vertical column as the Falcon in situ measurements in the BL in a coordinated manner, during 17 out of 17 and
17 out of 18 joint flights of the 2020 winter and summer campaigns, respectively. On the other hand, the CALIOP
instrument on CALIPSO satellite provides remote sensing measurements of aerosol extinction over the WNAO from
space, including three overpasses for which the two ACTIVAE aircraft performed under-flights during the 2020
summer campaign. In this section, we evaluate model simulated aerosol extinction profiles with HSRL-2 lidar and
CALIOP/CALIPSO satellite retrievals, characterize aerosol extinction vertical distribution, and examine potential
sources and processes responsible for aerosol extinction enhancements. Case studies of land-ocean horizontal gradient
of aerosol extinction, SNA and sea salt aerosol mixing, and long-range transport of the western U.S. fire smoke to the
ACTIVATE study domain in summer 2020, as observed by HSRL-2 and CALIOP, are also given.
**HSRL-2. Fig. 11** compares model simulated aerosol extinction profiles (550 nm, red lines) with aircraft HSRL-
2 lidar measurements (532 nm, black lines) averaged over 17 HSRL-2 flights during the 2020 winter and summer
campaigns, respectively. Also shown in the figure are results from model simulations with different injection heights
for BB emissions (**Table 1**). Hourly 3-D model output was sampled at the time and location of each HSRL-2
measurements from 17 (winter) and 17 (summer) flights, respectively. In winter, HSRL-2 observed high aerosol
extinction near the surface in the marine BL, which reflects North American BL outflow of aerosols to the WNAO
(section 4), and rapid decreases of aerosol extinction with increasing altitude. A layer of aerosol extinction
enhancement was observed at ~2.5 km altitude. The model largely overestimates extinction in the BL and
underestimates it in the free troposphere, resulting in a much sharper gradient between the BL and above (red line).





Note that on the other hand the model significantly underestimates monthly mean AOD relative to MODIS/Aqua
measurements in winter (**Fig. S3**). This discrepancy between HSRL-2 and MODIS measurements likely reflects the
sampling differences; for instance, many of the HSRL-2 flights occurred during cloudy conditions when MODIS
would not be able to measure AOD while HSRL-2 may still be able to measure aerosol extinction between gaps in
clouds. Increasing the injection height of BB emissions in the model improves the simulation of the vertical gradient
because of directly putting part of the emissions in the free troposphere. However, the model is not able to reproduce
the layer of aerosol extinction enhancement at ~2.5 km, suggesting vertical lifting is probably too weak in the model
in the wintertime. As shown below, this aerosol enhancement layer is also observed by CALIOP. In summer, the
standard model (red line) simulates the BL aerosol extinction reasonably well but fails to capture the large extinction
around 5 km as observed by HSRL-2. Increasing the injection height of BB emissions in the model to 0-5.5 km (green
line) or 2-10 km (blue line) results in much higher aerosol extinction at ~5 km, which is still lower than the mean
observed value by a factor of >2. However, the latter is weighted towards the extremely high aerosol extinction at ~5
km observed by HSRL-2 on Sep. 15 and 22, 2020, which will be discussed later in **Fig. 17**.
**CALIOP. Fig. 12a** (left column) compares model simulated aerosol extinction (550 nm) vertical profiles with
CALIOP measurements (532 nm) averaged over the period of Feb. 14 – Mar. 12, 2020, and for four subdomains of
the WNAO, respectively: North (39°-45°N, 72°-62°W), Central (32°-39°N, 78°-68°W), and South (25°-32°N, 82°-
72°W), as defined by Corral et al. (2020), and Bermuda (28°-35°N, 69°-59°W). For the North WNAO region, three
CALIOP data granules (2020-02-14T06-27-07ZN, 2020-03-03T06-21-54ZN, and 2020-03-04T17-37-24ZD) are
excluded due to cloud contamination. Examining aerosol extinction vertical profiles in the North and South WNAO
subdomains allows us to put the CALIOP observations in the Central WNAO, where all ACTIVATE 2020 research
flights occurred, in a context of latitudinal contrast. Model output averaged over 1-2 h LT and 13-14 h LT was sampled
at the date and location of nighttime and daytime CALIOP measurements, respectively. The right column shows the
model speciated aerosol extinction profiles corresponding to the sampled total aerosol extinction profile on the left
column. CALIOP observations show aerosols are mainly confined in the BL over the WNAO. In the Central WNAO,
the observed aerosol extinction reach a peak of ~0.05 km$^{-1}$ at ~0.5-1.0 km altitude, and exhibit layers of enhancements
in the free troposphere (e.g., ~2.5 km and ~4-6 km; note BL top < 1.6 km, **Fig. 2a**). The peak at ~2.5 km is consistent
with the HSRL-2 aerosol extinction enhancement at this altitude, as mentioned above (**Fig. 11**). The model captures
the decreasing trend of aerosol extinction with altitude but underestimates in the free troposphere where aerosol wet
scavenging is too fast in the standard model (section 5.2 and **Fig. 8**; this is also the case in the North WNAO
subdomain). Simulated speciated aerosol extinction values suggest sea salt contributes most to BL aerosol extinction
while SNA and OC contributions are comparable. In the North WNAO subdomain, CALIOP observed a similar
vertical gradient and the model suggests BL aerosol extinction are due to SNA aerosols more than OC and sea salt. In
the South WNAO subdomain, CALIOP observed the highest aerosol extinction of ~0.1 km$^{-1}$ (about 0.5 km above the
ocean surface) among the four subdomains. The model underestimates this peak by a factor of ~1.8, simulates a weaker
vertical gradient, and attributes BL aerosol extinction mostly to sea salt. It suggests that the model sea salt emissions
are biased low and/or sea salt is removed too fast in the model. This is in line with results from comparisons with





MODIS AOD (**section S1**, **Fig. S3**). In the proximity of Bermuda, vertical aerosol extinction is more concentrated
towards the surface due to the dominant contribution from sea salt.
While summertime aerosols are mostly confined to the BL (top < 1 km), a larger fraction of the total AOD is
contributed by aerosols in the free troposphere compared to winter (Corral et al., 2020; **Fig. 12b**). This is evidenced
by layers of aerosol extinction enhancements, especially between ~3-8 km in the North WNAO subdomain. The
CALIOP aerosol extinction values between ~3-6 km in the Central WNAO are not as high as those from King Air
HSRL-2 measurements (right panel, **Fig. 11**), presumably because of the spatiotemporal mismatch between the
CALIPSO satellite overpass and King Air HSRL-2 sampling. The CALIOP-observed aerosol extinction peak (~0.08
km$^{-1}$) at ~0.5 km altitude in the Central WNAO is ~60% higher than in winter, consistent with the model result of
more sea salt in summer (**Fig. 12ab**; also see section 5.1). The standard model simulation with BB emissions injected
within the BL failed to capture the smoke aerosol extinction enhancements between ~3-8 km in the North WNAO;
using the 0-5.5 km injection heights clearly improves the simulation of smoke aerosol extinction, although the altitude
of the simulated smoke plume is somewhat lower than the observed plume height (bottom panels, **Fig. 12b**). Smoke
aerosol extinction is mainly contributed by organic aerosols, as suggested by the model. In contrast to winter, sea salt
aerosols make dominant contributions to the BL total aerosol extinction not only in the Central and South WNAO and
Bermuda but also in the North WNAO subdomain, which reflects the southwesterly transport of low-latitude marine
air along the WNAO as part of the Bermuda High in summer. SNA and OC contributions are also important in all
four subdomains while dust contributions are significant at lower latitudes, especially Bermuda.
**Land-Ocean aerosol extinction gradient.** We present a case where a large horizontal gradient of aerosol
extinction from the eastern U.S. coast going eastwards was observed by HSRL-2. **Fig. 13a** shows model simulated
hourly total aerosol extinction at ~1 km altitude over the WNAO during the King Air morning flight (14-17 UTC) on
March 12, 2020. Gray lines indicate the complete flight track with overlayed red lines denoting the flight tracks for
each hour. On the morning of March 12, King Air conducted a statistical survey flight as well as an ASTER/Terra
under-flight. It flew eastwards from LaRC and turned northeastward near the ZIBUT waypoint, followed by flying
back via ZIBUT after conducting an ASTER under-flight along ~69.7°W. The entire flight took about 3 hours (14:00-
17:00 UTC). **Fig. 13b** compares the time-height cross-section of aerosol extinction observed by the King Air HSRL-
2 lidar with those of model total and speciated aerosol extinction during the flight. **Fig. 14a** shows latitude-height
cross-section of aerosol extinction measured by CALIOP (532 nm) compared to that of model aerosol extinction (550
nm) over the WNAO at 7:08 UTC (3:08 am LT, about 7 hours before the King Air flight), Mar. 12, 2020. **Fig. 14b**
shows the corresponding curtain plots for model speciated aerosol extinction, as well as accumulated sea salt (SSa),
coarse-mode sea salt (SSc), RH, cloud fraction, and effective cloud extinction.
HSRL-2 observed very high aerosol extinction in the BL over land and off the coast with a decreasing trend
toward the marine region. The model captures this trend (**Fig. 13a**) but generally overestimates BL aerosol extinction
(**Fig. S5**). Despite HSRL-2 being unable to obtain meaningful retrievals in the BL for a large portion of the flight track
because of the interference of low-level clouds, the observed general pattern of BL aerosol extinction over land and
ocean is very similar to the model result (**Fig. 13b**). A thin layer of aerosol at 2-3 km seen by HSRL-2 is missing in
the standard model simulation. This is likely due to the low injection heights (within BL) of southeast U.S. BB




emissions in the model. GEOS-Chem speciated aerosol extinction suggest that SNA and OC are the main contributions
over land and off the coast with a similar magnitude of extinction (**Fig. 13b**). Over the ocean (~14:30-16:30 UTC),
SNA, OC, and SS all contribute to the thin aerosol layer close to sea surface with a slightly higher contribution from
SNA. The CALIPSO descending swath scanned the WNAO region about seven hours earlier and intersected almost
perpendicularly the flight track (**Fig. 14a**). However, the CALIOP retrieved only low aerosol extinction between 32-
40°N (note ~37°N is the King Air flight latitude at 14:00 UTC). This is because the CALIOP laser signal was largely
attenuated by the presence of optically thick clouds at these latitudes (**Fig. 14b**). Indeed, the corresponding model
results show high aerosol extinction in the BL along the CALIPSO orbit track. Model simulated speciated aerosol
extinction suggest SNA and OC contribute comparably to aerosol extinction at ~37°N while sea salt becomes more
important towards lower latitudes (**Fig. 14b**).
**SNA and sea salt mixing.** We present a case where SNA and sea salt aerosols are mixed as suggested by our
model analysis of HSRL-2 and CALIOP observations. **Fig. 15ab** presents plots similar to **Fig. 14ab** but for the King
Air flight on March 6, 2020, when the King Air conducted a statistical survey flight to the OXANA waypoint and
then to a southwest point (32.8°N, 75.2°W), and encountered a high aerosol layer in the marine BL (**Fig. 15a**). The
aircraft returned along the same flight track. HSRL-2 observed enhanced aerosol extinction in the lower troposphere
despite missing retrievals for most of the flight period (~18:00-22:00 UTC) due to attenuation by widespread marine
stratus clouds (GOES visible image, not shown). The model attributes the observed enhanced aerosol extinction in the
BL at ~20:00 UTC to sea salt mixed with SNA aerosols (**Fig. 15b**). The detachment of the SNA aerosol extinction
layer from the sea surface suggests its major source is U.S. continental anthropogenic emissions. GEOS-Chem also
simulates high aerosol extinction over the south WNAO region centered near the returning point of the flight.
CALIPSO overpassed the same region at the ascending node on the next day, Mar. 7 (**Fig. 16a**). CALIOP
measurements (532 nm) over the WNAO at 17:53 UTC that day show enhanced aerosol extinction in the lower
troposphere (< ~2.5 km) south of 39°N. The model reproduces this enhancement (**Fig. 16a**) and attributes it mainly to
coarse-mode sea salt (<32°N) and SNA (>32°N) (**Fig. 16b**). The latter is thus consistent with the model analysis of
HSRL-2 measurements on Mar. 6 (**Fig. 15b**). It is interesting to note that at < 35°N, especially lower latitudes, the
model simulates very high sea salt (mostly coarse-mode) aerosol extinction near the top of the marine BL where RH
and cloud extinction are high. This feature is typically seen over the WNAO in both CALIOP aerosol extinction
profiles and GEOS-Chem simulations. It suggests strong hygroscopic growth of sea salt particles and sea salt seeding
of marine BL clouds.
**Transport of the western U.S. fire smoke.** The above evaluation of model simulations with HSRL-2 and
CALIOP composite aerosol extinction observations in summer 2020 has suggested that using 0-5.5 km BB emission
injection heights significantly improves model performance in simulating the transport of the western U.S. fire smoke
plumes to the WNAO (**Fig. 11**, **Fig. 12b**). We present here model simulations of HSRL-2 aerosol extinction
measurements with large enhancements in the free troposphere from several individual flights to further demonstrate
the importance of BB emission injection heights with respect to the overall model performance. **Fig. 17** compares
time-height cross-sections of HSRL-2 lidar aerosol extinction (532 nm, left column) with those of model aerosol
extinction (550 nm, middle and right columns) for the flights of Aug. 26, Aug. 28, Sep. 15, and Sep. 22, respectively,



2020. Model results from the standard simulation (BB emissions injected into the BL) and the "BB0-5.5km"
simulation (BB emissions injected into 0-5.5 km) as listed in **Table 1** are shown. In the case of Aug. 26, the aerosol
plume at ~1.5-2 km altitude was missing in the standard simulation but captured by BB0-5.5km. In the case of Aug.
28, both simulations perform similarly and reproduce the aerosol extinction enhancements in the free troposphere.
Plumes with very high aerosol extinction were observed by HSRL-2 between 4-6 km on Sep. 15, and Sep. 22. While
the standard model failed to simulate these smoke plumes on both days, the BB0-5.5km simulation successfully
captures the transported smoke plumes albeit with much lower aerosol extinction.

We conduct a case study of long-range transport of the western U.S. fire smoke to the WNAO on Sep. 23, 2020,
which was captured by both CALIOP and under-flying ACTIVATE aircraft. **Fig. 18** shows time-height cross-section
of HSRL-2 aerosol extinction (532 nm) compared to that of model total (and speciated) aerosol extinction (550 nm)
for the King Air flight in the afternoon (~17-20 UTC) of that day. Also shown are the GOES-16 visible image (18:21
UTC; NASA Langley SATCORPS group) superimposed with King Air (red) and Falcon (yellow) flight tracks. HSRL-
2 observed a layer of very high aerosol extinction between 1.5-4.0 km during 17:00-18:00 UTC and another layer of
enhanced aerosol extinction between 1.5–3 km during ~18:30-20:00 UTC. The model simulated total aerosol
extinction shows a similar layer of aerosols (higher extinction between 17:00-18:30 UTC) during the entire flight, but
it is consistently thinner (between ~1.7-3.5 km) and located at a slightly lower altitude. Using the 0-5.5 km BB
emission injection heights does not correct this model bias (not shown). The model simulated speciated aerosol
extinction suggest that the dominant contribution to the high aerosol extinction layer is from OC with a small
contribution from SNA aerosols. The clear isolation above the marine BL indicates that the aerosols are very likely
from long-range transport in the free troposphere. Model results show a sea salt extinction component in the marine
BL between ~18:00-19:00 UTC as the aircraft was at the farthest location from the coast. Unfortunately, the HSRL-2
retrievals are not available for that time window due to cloud interference but do show some BL aerosols before and
after the data gap around 18:00 UTC.

In situ measurements from Falcon flying under King Air also show high CO (~200 ppbv) and aerosols (mostly
organics) between ~2-4 km during the flight on Sep. 23, 2020 (**Fig. S6**). Although the model failed to reproduce the
high CO concentrations in this layer (also see section 5.2), it simulates relatively weak CO enhancements due to BB
near the bottom (~2-3 km) of the observed layer. Similarly, the model puts the BB aerosol plume at an altitude about
1 km lower than that observed. Injecting BB emissions to higher altitudes has little effect on the simulated plume
altitude in this case. As observed, the simulated BB plume is mainly composed of organic aerosols.

Back trajectories were calculated to determine the origin of the aforementioned smoke plume. **Fig. 19ab** shows
the FLEXPART simulated upwind residence times for the air masses arriving at the location (latitude 36.87°N,
longitude 72.57°W, altitude ~3.5 km) at 17:13 UTC, Sep. 23, 2020. A major part of the air masses originated from the
BL of the western U.S. about 3-4 days upwind**,** where large fire events occurred during early and mid-September as
depicted by the QFED BB CO emission inventory (**Fig. 19c**). The trajectories in conjunction with surface weather
maps (**Fig. 19d**; https://www.wpc.ncep.noaa.gov/dailywxmap/index_20200920.html) suggest that the air masses
subsided to within the BL behind an eastward-moving cold front over mid-western states (~110°W-105°W) during



Sep. 19-20 before being lifted to the free troposphere during Sep. 20-21, followed by fast transport between 3.5-5 km
altitudes to the ACTIVATE domain.
While both ACTIVATE aircraft captured the smoke plume from the western U.S. fires, the satellite
measurements from CALIPSO can put it in a context of latitudinal extent. **Fig. 20** (left column) shows the curtain of
aerosol extinction measured by CALIOP over the WNAO at 17:45 UTC (1:45 pm LT), Sep. 23, 2020, in comparison
with model results from both the standard and BB0-5.5km simulations. Also shown (right column) is the comparison
corresponding to the CALIPSO nighttime overpass at 07:08 UTC (03:08 am LT) when the noise in the data is smaller
compared to daytime data. During daytime, CALIOP observed high aerosol extinction in the lower troposphere at 27-
33°N and 41-45°N, between 2.5-4.0 km altitudes at 35-37°N, and around 4.5 km altitude at 27-31°N. The altitude 2.5-
4.0 km of the aerosol plume at 35-37°N, where the CALIPSO under-flight occurred, is close to the plume altitude
~1.5-4.0 km observed by HSRL-2 during 16:94-17:48 UTC (**Fig. 18b**). The standard model reasonably reproduces
the general pattern of aerosol extinction as a function of latitude and altitude, but underestimates it in the lower
troposphere around 43°N and misses aerosols observed at 4.5 km between 27-31°N. For the latter, the BB0-5.5km
simulation increases aerosol extinction but the simulated plume altitude is still ~ 1 km lower. The CALIOP nighttime
observations made about 10 hours earlier in close proximity show a more coherent pattern of the latitude-altitude
distribution, i.e., enhanced aerosol extinction in the BL between 25-43°N and distinct aerosol plumes in the free
troposphere (2.5-5.5 km, 35-43°N; 4-5 km, 27-33°N), with missing retrievals between 43-49°N. The model simulates
coherent aerosol plumes in the free troposphere at a wider latitude range (35-49°N), especially with the BB0-5.5km
simulation. BB0-5.5km suggests that the CALIOP observed plume at 4.5 km between 27-33°N is a southward
extension of the smoke plume at higher latitudes. The simulated aerosol plume is nevertheless at least 0.5 km lower
in altitude than that observed by CALIOP.
The weaker vertical transport of trace gases and aerosols in the model is likely a result of two model uncertainties:
remapping of meteorological data from the native cubed-sphere grid of the parent GEOS-5 GCM to an equally
rectilinear (latitude-longitude) grid, and degradation of the spatial and temporal resolutions of the input meteorological
data (Yu et al., 2018). Yu et al. (2018) showed that the remapping and the use of 3-hourly averaged wind archives
may lead to 5-20% error in the vertical transport of a surface-emitted tracer $^{222}$Rn in offline GEOS-Chem simulations
compared to online GEOS-5 simulations. It was attributed partly to the loss of organized vertical motions. Degrading
the spatial resolution of the meteorological data for input to GEOS-Chem (e.g., $2° \times 2.5°$ used in this study) further
weakened vertical transport because organized vertical motions are averaged out at a coarser resolution. Such
inefficient vertical transport in coarse-resolution GEOS-Chem was also noted previously in the simulations of Asian
Tropopause Aerosol Layer (Fairlie et al., 2020) and upper-tropospheric $^{222}$Rn (Zhang et al., 2021), as constrained by
observations. Restoring the lost vertical transport by implementing the modified Relaxed Arakawa–Schubert
convection scheme in GEOS-Chem would alleviate this issue (He et al., 2019). On the other hand, the spatial resolution
for global models may be too coarse to resolve mean vertical motion that can be better resolved by regional models,
as illustrated by Fast et al. (2016) when simulating the observed aerosol layers transported from North America over
the Atlantic Ocean.





## 6. Source attributions of AODs over the WNAO during ACTIVATE

In this section, we quantify the contributions of different emission types to the AODs over the eastern U.S. and WNAO during ACTIVATE in the model. **Fig. 21** shows the absolute and percentage changes in the average AODs for the periods of the 2020 winter (Feb.14-Mar.12) and summer (Aug.13-Sep.30) campaigns, respectively, when anthropogenic (including biofuel), BB, biogenic, marine, and dust emissions are separately turned off in the model. In winter, anthropogenic emissions make dominant contributions (70-90%) over land between 36-48°N, 30-60% in the N box region, 20-40% in the S box region, with a decreasing contribution trend from NW to SE. At Bermuda, about 20% of AOD is due to anthropogenic emissions. BB emissions make the largest contributions (up to 40-50%) in southeastern U.S. coast and contribute about 10-30% of AODs in the N+S box regions. Marine emissions (mainly sea salt) contribute 30-60% over most of the N+S box regions, with an increasing contribution trend from NW to SE, which is opposite of the trend of anthropogenic emission contributions. Biogenic and dust emissions make only small contributions to AODs throughout the domain. In summer, anthropogenic emission contributions are reduced over both land and WNAO compared to winter, with 20-30% contribution in the N+S box regions. By contrast, BB emission contributions increase substantially over land and WNAO relative to winter, reflecting the influences of the western U.S. fire smoke as well as BB emissions from the southeast U.S. during summer. Interestingly, there is an apparent pathway for transport of smoke plumes towards Bermuda (left second row, **Fig. 21b**) as also demonstrated in recent work (Aldhaif et al., 2021; Mardi et al., 2021); this is consistent with the smoke AOD events observed by AERONET at Tudor Hill, Bermuda (section 5.3). At Bermuda, BB emissions contribute more to AOD than anthropogenic emissions in summer (~20% vs. ~15%). Biogenic emission contributions (10-20%) are mostly confined in the southeast U.S. Marine emission contributions to AOD in the N+S box regions and around Bermuda are significantly higher compared to winter (left fourth row, **Fig. 21b**), despite higher marine emission contributions south of 30°N in winter. African dust contributions to AOD (>10%) are seen mainly to the south of Bermuda but extend as far as Florida and the Gulf of Mexico.

## 7. Summary and conclusions

We have simulated tropospheric aerosols over the western North Atlantic Ocean (WNAO) during the winter (Feb. 14 - Mar. 12) and summer (Aug. 13 – Sep. 30) campaigns of the NASA Earth-Venture Suborbital-3 ACTIVATE 2020 mission (Sorooshian et al., 2019, 2023) using the GEOS-Chem global chemical transport model driven by the MERRA-2 assimilated meteorology (at 2°×2.5° horizontal resolution). Model results are evaluated with measurements from two aircraft, the low-flying HU-25 Falcon and high-flying King Air, as well as ground-based and satellite observations. Our objective is to characterize and improve understanding of the aerosol life cycle, transport, and distribution over the WNAO during the two campaigns. Major work and results are summarized below.

1) Contrasting atmospheric circulation patterns and meteorological conditions were prevalent during Feb.-Mar. and Aug.-Sep. ACTIVATE 2020. In winter, the flights took place in a recurring location for the passages of cold fronts driven by the Northern Hemisphere mid-latitude cyclones and sampled the North American continental outflow extensively. Flights to the north often occurred during post-frontal conditions to sample and study MBL clouds, especially during cold air outbreaks. The southern flight region experienced stronger uplifting and precipitation





scavenging. In summer, the strengthened Bermuda High extended westward with southwesterly winds offshore along
the U.S. SE coast (~32-36°N) and easterly in the subtropics (<30°N). Compared to winter, the horizontal gradients in
surface temperature, RH, vertical pressure velocity, precipitation, and PBL height were much weaker, and convective
precipitation was much stronger in the flight domain.
2) Transport in the BL behind cold fronts is a major mechanism for the North American pollution outflow to the
WNAO. The winter campaign encountered about four major periods with cold fronts passing through the study area,
during which continental pollution (e.g., CO) was swept in the BL southeastward to the ACTIVATE flight latitudes
(32.4-39.8°N). In summer, intrusion of cold air deep into the flight domain did not occur until late September since
midlatitude cyclones were shifted northward.
3) Major pathways for vertical transport of trace gases and aerosols over the North American continental outflow
region include uplifting ahead of cold fronts, convective transport, and BL turbulent mixing. In winter, large-scale
vertical transport ahead of cold fronts was the dominant process responsible for lifting CO out of the BL over the U.S.
East Coast and the flight domain, followed by rapid eastward transport in the free troposphere. Convection was also
important, especially in the southern flight area, but became even more important in summer. By contrast, BL turbulent
mixing was found to be the dominant process responsible for the upward transport of sea salt within and ventilation
out of the BL over the flight domain in winter, while both convection and turbulent mixing were important in uplifting
sea salt to the free troposphere in summer.
4) We characterized the model simulated aerosol species with respect to their distributions, mass loadings, and
optical depths (AODs) over the WNAO. In winter, the horizontal distributions of SNA, BC, POA, and SOA
concentrations in the BL largely reflected the frequent transport of continental pollution behind cold fronts. Sea salt
had the largest mass among aerosols over the WNAO, with maximum in the easterly trade wind region and to the east
of the northern flight domain. BC, SOA, and dust abundances across the flight domain were relatively small. In
summer, BL sulfate concentrations were significantly higher, resulting from stronger production from oxidation of
$SO_2$. BL nitrate concentrations decreased substantially due to the volatility of ammonium nitrate at higher temperature
and more sulfate competing with nitrate for ammonium. Substantially higher SOA concentrations reflect the large
production from strong oxidation of VOCs. The vertical extent of the major North American continental outflow
aerosols was significantly higher because of the impact of convective lifting. BL sea salt abundance increased in
summer over the WNAO, especially from the ACTIVATE flight areas to Bermuda, because of stronger winds. Dust
amounts also significantly increased due to long-range transport of dust emissions from North Africa. In both seasons,
sea salt, OA, and SNA were the main contributors to the mean total AOD in the flight areas. The strong hygroscopic
growth of fine aerosols results in a much larger fraction of AOD (versus aerosol mass) from SNA (or OA).
5) We evaluated model simulated vertical profiles of CO, SNA, and OA concentrations with Falcon aircraft
AMS measurements and performed sensitivity experiments to quantify their sources. In winter, outflow of pollution
from continental sources dominated the lower troposphere, causing a sharp vertical gradient in CO, SNA, and OA
concentrations at ~1.0 km altitude; in summer, impacts of convection and BB sources increased and those gradients
were weaker. Extending the BB emission injection height from within the BL to 0-5.5 km largely improved or
corrected the model low biases in simulated aerosol enhancements in the free troposphere during the summer





campaign. SNA aerosols are predominantly from continental anthropogenic emissions, but summertime BB
contributions are also important between ~3-6 km. OA in the free troposphere are mainly from BB whereas those in
the marine BL have sources from BB, anthropogenic, and marine emissions.
6) Intensive aerosol profile measurements from ACTIVATE 2020 provide useful constraints on model aerosol
scavenging due to stratiform precipitation. Uncertainty in CWC used in GEOS-Chem has a large impact on the
simulated tropospheric aerosols over the ACTIVATE study domain. Using the MERRA-2's spatiotemporally varying
CWC (versus a fixed value) improves model simulations of BL aerosol (especially sulfate) concentrations and
AERONET AODs in the domain in winter. However, this approach leads to too fast scavenging in the free troposphere.
The model also had some difficulties in reproducing the surface aerosol concentration and deposition flux
measurements in the eastern U.S. coastal region as well as AOD retrievals from MODIS/Aqua satellite measurements
over the WNAO. Fully implementing the revised wet scavenging scheme of Luo et al. (2020) in the model could
improve the model performance.
7) We evaluated model simulated aerosol extinction (at 550 nm) profiles with King Air HSRL-2 lidar and
CALIOP/CALIPSO satellite retrievals (at 532 nm) during the two campaigns. In winter, HSRL-2 observed high
aerosol extinction in the marine BL associated with the North American continental outflow. A layer of aerosol
extinction enhancements was observed at ~2.5 km altitude. The model simulates a much sharper gradient compared
to HSRL-2 between the BL and free troposphere, suggesting the vertical lifting is probably too weak in the model. In
summer, HSRL-2 observed much higher aerosol extinction in the BL and large extinction enhancements around 5.0
km altitude. The standard model fails to capture the latter but can be improved by using higher BB emission injection
heights.
8) In winter, the CALIOP aerosol extinction over the central WNAO reached a peak at ~0.5-1.0 km altitude and
showed layers of enhancements in the free troposphere (e.g., the peak at ~2.5 km altitude also observed by HSRL-2).
The model captures the vertical trend of aerosol extinction but underestimates extinction in the free troposphere largely
due to too fast scavenging. SNA aerosols contribute the most in the BL over the North WNAO while sea salt
predominates over the southern WNAO. For the latter, the model substantially underestimates aerosol extinction in
the lower BL, suggesting too low sea salt emissions or too fast removal. In summer, free tropospheric aerosols
contribute a larger fraction of AOD relative to winter (Corral et al., 2020), The significantly higher extinction peak
observed in the BL over the central WNAO compared to winter is consistent with simulated higher sea salt in summer.
Injecting BB emissions between 0-5.5 km altitudes apparently improves the simulation of smoke aerosol extinction,
especially in the northern WNAO subdomain. A feature typically seen over the WNAO (< 35°N) as suggested by
CALIOP aerosol extinction profiles and GEOS-Chem simulations is very high sea salt (mostly coarse-mode) aerosol
extinction near the top of the marine BL where RH and cloud extinction are high. The latter suggests strong
hygroscopic growth of sea salt particles and sea salt seeding of marine stratus clouds.
9) We conducted a case study of long-range transport of the western U.S. fire smoke to the WNAO on Sep. 23,
2020, which was captured by both CALIOP and the under-flying ACTIVATE aircraft. The CALIPSO measurements
allowed us to put this smoke transport event in a context of latitudinal extent. Model simulations of HSRL-2 aerosol
extinction measurements with large enhancements in the free troposphere from several individual flights (Aug. 26,



Aug. 28, Sep. 15, and Sep. 22, 2020) demonstrate that injecting BB emissions into 0-5.5 km altitudes often improves
the model performance. Case studies also show that the model reasonably captures the continental outflow of aerosols,
land-ocean aerosol extinction gradient, and mixing of anthropogenic aerosols with sea salt.

10) We quantified the contributions of different emission types (anthropogenic, BB, biogenic, marine, dust) to

the AOD over the eastern U.S. and WNAO in the model. In winter, anthropogenic emission contributions dominate
near the coast and decrease southeastward. BB emissions contribute most to AOD in southeastern U.S. coast and
account for ~10-30% of AOD in the flight area, while marine emissions contribute 30-60% over most of the flight
area southeastward. In summer, anthropogenic emission contributions to AOD are reduced but BB emission
contributions increase substantially. An apparent pathway for transport of smoke plumes towards Bermuda is
identified (Aldhaif et al., 2021; Mardi et al., 2021) and is consistent with the smoke AOD events observed by
AERONET at Tudor Hill, Bermuda. BB emissions contribute more to AOD at Bermuda than anthropogenic emissions
in summer (~20% vs. ~15%). Biogenic emission contributions (10-20%) are mostly confined in the southeast U.S.
Marine emission contributions to AOD in the flight area and around Bermuda are significantly higher relative to
winter. African dust contributions to AOD (>10%) are seen mainly to the south of Bermuda but extend as far as Florida
and the Gulf of Mexico.

The above results on aerosol lifecycle, transport, and distribution have important implications for studies of

aerosol-cloud-meteorology interaction during ACTIVATE 2020. For instance, transport of continental aerosols over
the WNAO may modulate cloud microphysics and precipitation. Recently, Painemal et al. (2023) analyzed wintertime
BL cloud synoptic variability over the WNAO and linked the occurrence of a maximum in cloud droplet number
concentration with continental aerosols during cold air outbreaks. Correctly representing aerosol distribution and
variability is thus critical in simulating aerosol indirect effects on clouds. Biomass burning aerosols can affect the
whole troposphere and interact with clouds directly or indirectly, as suggested by a case study of smoke transport from
the western U.S. during ACTIVATE on Aug. 26, 2020 (Mardi et al., 2021). Mardi et al. also associated BB days with
higher cloud drop number concentrations and lower drop effective radius. Our work implies that using reasonable BB
emission injection heights in global models, among other factors, plays an essential role in representing smoke-cloud
interactions. The high coarse-mode sea salt aerosol extinction along with high RH and cloud extinction near the top
of the marine BL over the WNAO (< 35°N), as identified in this work, suggests a potential ideal region for studying
giant CCN – cloud interactions (Gonzalez et al., 2022).

This study highlights the following areas for recommended future work to improve the modeling and

understanding of tropospheric aerosol life cycle, transport, and distribution over the WNAO. An evaluation of the
MERRA-2 CWC, including its partition between liquid and ice water in the vertical column, with available aircraft
and satellite observations is required for a better representation of aerosol scavenging in GEOS-Chem. The liquid-ice
partitioning affects the scavenging efficiencies of aerosols due to both warm and ice clouds. Accurate BB emission
injection heights derived from daily or hourly observations from space (e.g., from the TEMPO geostationary satellite)
are expected to significantly enhance the model's capability to simulate smoke aerosols and their vertical distribution
over North America and the WNAO. Furthermore, efforts to improve parameterizations of sea salt emissions for use



in global models should be encouraged. Inefficient vertical transport in coarse-resolution models may also be
improved by using high-resolution and/or regional models.

*Data availability.* Observational data for model evaluation are introduced in Section 3. All data from the Falcon and
King Air are publicly archived on NASA Atmospheric Science Data Center (ASDC)'s Distributed Active Archive
Center (DAAC) and accessible at https://doi.org/10.5067/SUBORBITAL/ACTIVATE/DATA001. GEOS-Chem code
v11-01 used in this work is available at https://zenodo.org/records/10982278 (Liu and Zhang, 2024).

*Supplement.* The supplement related to this article is available online at: https://doi.org/XX.XXX/XXX-supplement
(TBD).

*Competing interests.* At least one of the (co-)authors is a member of the editorial board of Atmospheric Chemistry
and Physics.

*Author contributions.* Conceptualization: HL, BZ, RM, and AS. Model simulation: HL and BZ. Analysis and initial
draft preparation: HL and BZ with contributions from HC. Data collection: RM, LZ, RF, AS, MS, AJS, JH, EC, MF,
TS, CH, GD, JN, JD, and YC. CALIOP data analysis: HC, JT, and MV. Manuscript review, comments, and editing:
all authors. Model modification and improvement: HL, BZ, GL, FY, CK, and MJ.

*Acknowledgements.* The work was funded by ACTIVATE, a NASA Earth Venture Suborbital-3 (EVS-3) investigation
funded by NASA's Earth Science Division and managed through the Earth System Science Pathfinder Program Office.
HL and BZ acknowledge the partial support of the NASA EVS-2 NAAMES mission for model development. The
GEOS-Chem model is managed by the Atmospheric Chemistry Modeling Group at Harvard University with support
from NASA ACMAP and MAP programs. The GEOS-Chem support team at Harvard University and Washington
University at Saint Louis (WashU) is acknowledged for their effort. GEOS-Chem input files were obtained from the
GEOS-Chem Data Portal enabled by WashU with assistance from Andrew Schuh of Colorado State University. HL
would like to thank Brett Gantt (EPA) for assisting with producing the MODIS/Aqua chlorophyll-$\alpha$ data, Lee Murray
(University of Rochester) for providing the lightning $NO_x$ emission file for use with GEOS-Chem, and Robert Levy
(NASA) for helpful discussions on the use of MODIS aerosol data. NASA Center for Climate Simulation (NCCS)
provided supercomputing resources. PIs and staff for the three AERONET sites are acknowledged for their effort. The
Pacific Northwest National Laboratory (PNNL) is operated for DOE by Battelle Memorial Institute under contract
DE-AC05-76RLO1830.

*Financial support.* This research was supported by NASA grant 80NSSC19K0389 in support of the ACTIVATE
mission.




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





**Tables and Figures**


**Table 1.** GEOS-Chem model simulations for Feb.-Mar. and Aug.-Sep. 2020.

| Model simulations | Stratiform cloud water content (CWC)* | QFED biomass burning emission injection heights | Emissions |
|---|---|---|---|
| standard | MERRA-2* | PBL | all |
| fixedCWC | $1.0 \times 10^{-3}$ kg m$^{-3*}$ | PBL | all |
| BB0-5.5km | MERRA-2 | 0-5.5 km | all |
| BB2-10km | MERRA-2 | 2-10 km$^{+}$ | all |
| noanthbf | MERRA-2 | PBL | zero anthropogenic and biofuel emissions |
| nobb | MERRA-2 | N/A | zero biomass burning emissions |
| nobg | MERRA-2 | PBL | zero biogenic emissions |
| nomari | MERRA-2 | PBL | zero marine emissions |
| nodu | MERRA-2 | PBL | zero dust emissions |

*CWC is used in the model parameterization for aerosol scavenging due to stratiform precipitation. Its value is either
taken from MERRA-2 (Luo et al., 2019, 2020) or assumed a fixed constant of $1.0 \times 10^{-3}$ kg m$^{-3}$ (Del Genio et al., 1996;
Wang et al., 2011).
$^{+}$This simulation is for Aug.-Sep. 2020 only.




1410

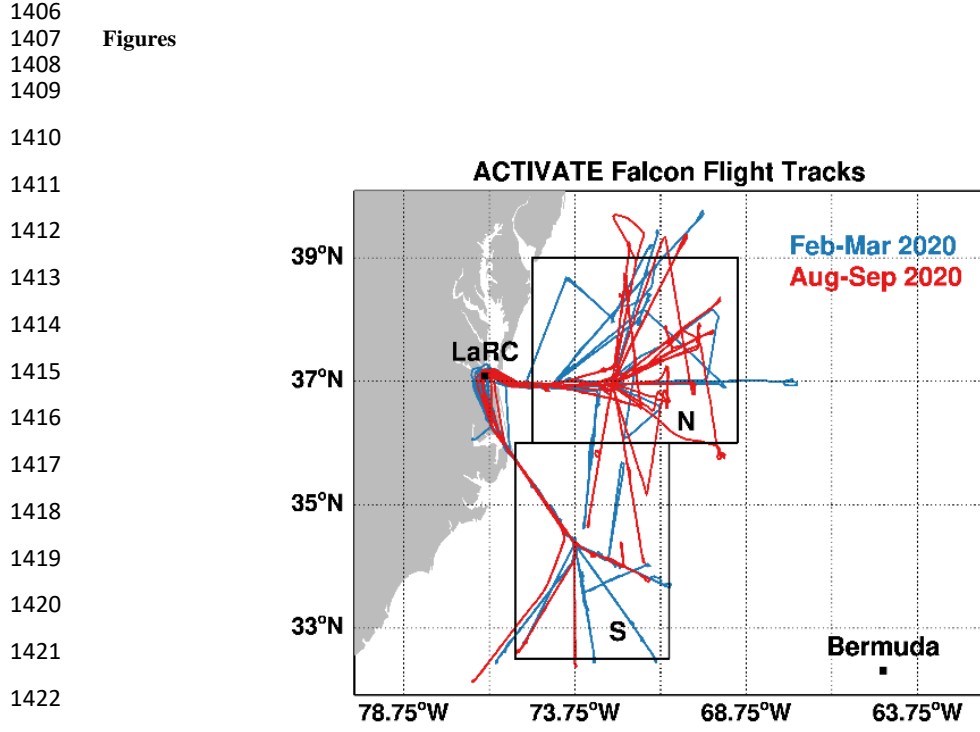

1423

**Figure 1.** Falcon (HU-25) flight tracks during the winter (Feb. 14 - Mar. 12) and summer (Aug. 13 – Sep. 30) campaigns of
ACTIVATE 2020. Almost all flights are based out of NASA Langley Research Center (LaRC), with a few in the winter campaign
based out of the nearby Newport News/Williamsburg International Airport. The aircraft sampling domain is divided at 36°N into
two box regions, the north ("N"; 36-39°N, 69-75°W) and the south ("S"; 32.5-36°N, 71-75.5°W), for data analysis.

1428

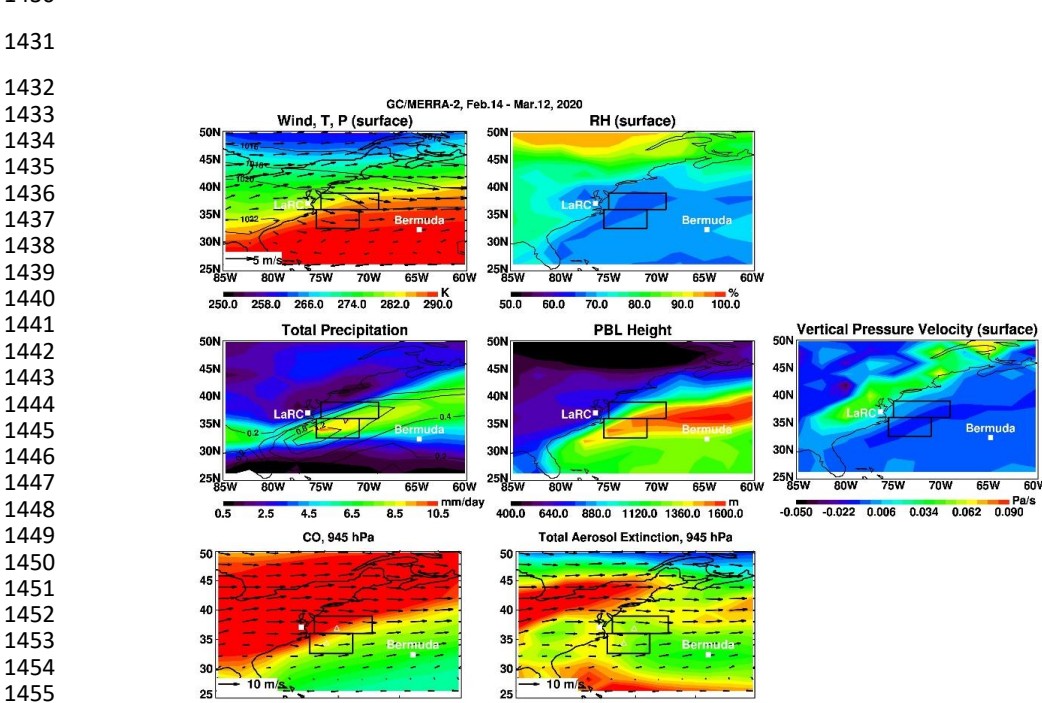

**Figure 2a.** Mean meteorological conditions from MERRA-2 and near-surface (945 hPa) CO concentrations (ppbv) and aerosol extinctions at 550 nm (km$^{-1}$) in GEOS-Chem during Feb.-Mar. 2020. **Top row**: Left panel shows horizontal wind (m/s, vectors) and temperature (K, image) at the model bottom layer, as well as sea level pressure (hPa, contours). Right two panels show relative humidity (RH, %)) and vertical pressure velocity ($\omega$, Pa/s). **Middle row**: Total precipitation (mm/day, image), convective precipitation (mm/day, contours), and PBL height (m). **Bottom row**: Model simulated CO concentrations and aerosol extinction overlayed with horizontal wind vectors at 945 hPa. The two rectangular boxes denote major flight areas (see "N" and "S" in **Fig. 1**) of Feb.-Mar. and Aug.-Sep. 2020. The locations of LaRC and Bermuda are marked by white squares.



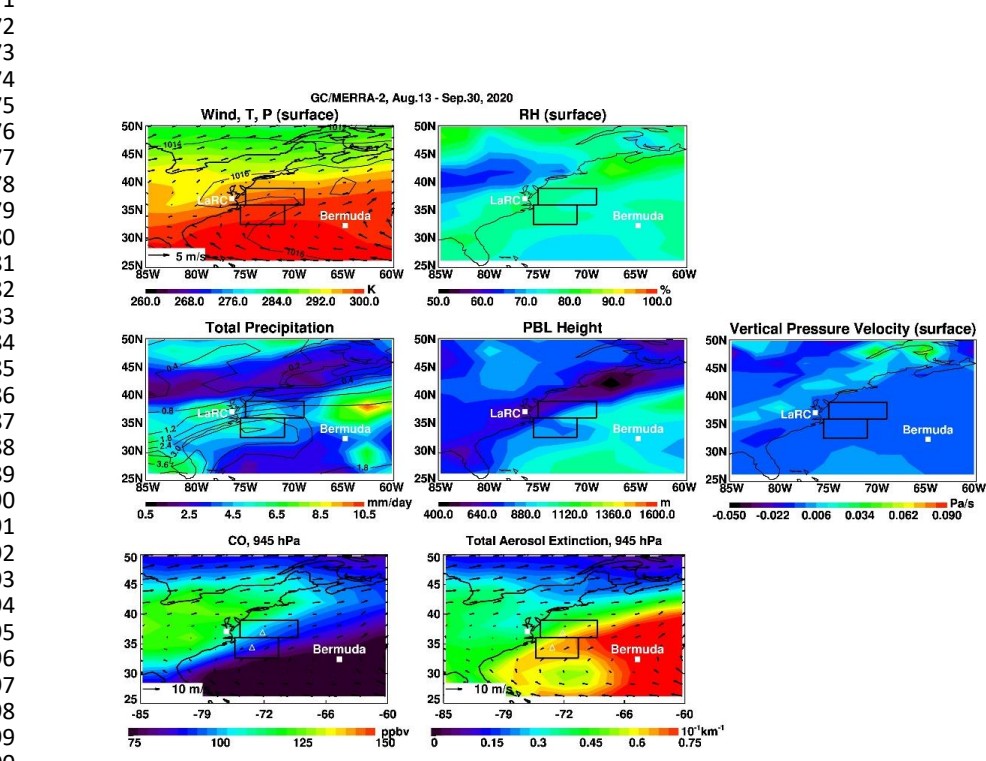

**Figure 2b.** Same as Fig. 2a, but for Aug.-Sep. 2020 (with different colorbars for temperature and total aerosol extinction).



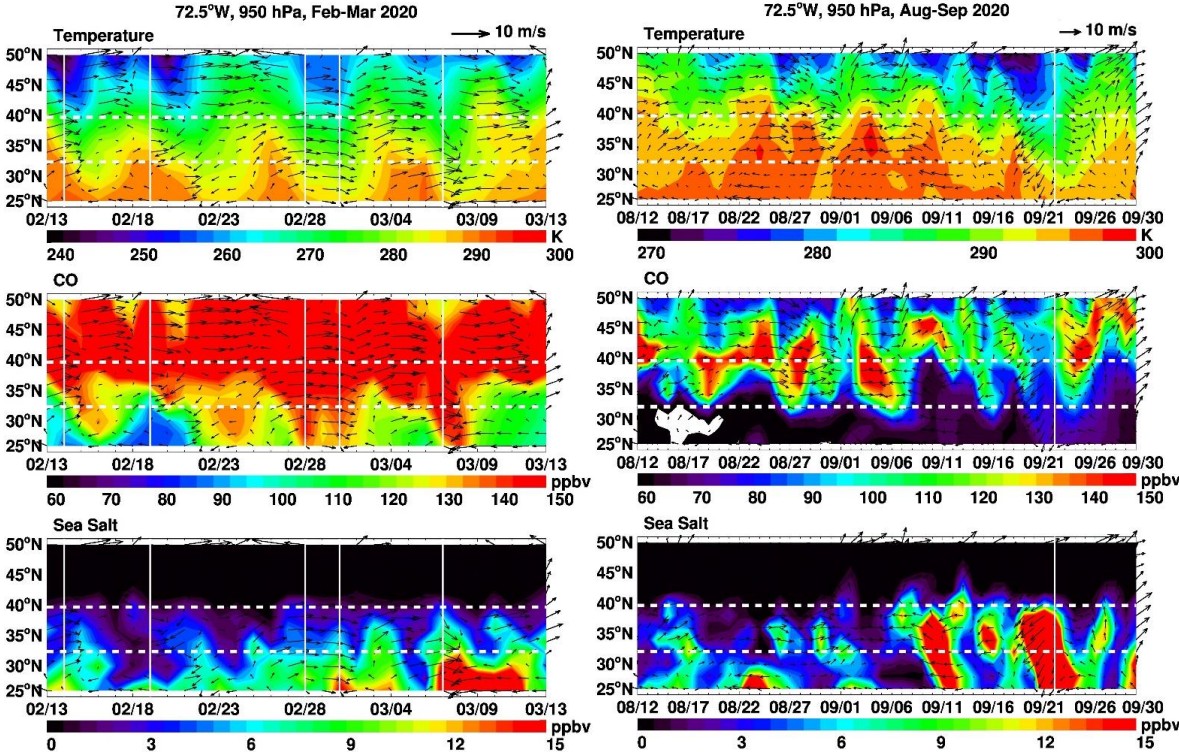

**Figure 3.** Spatiotemporal evolution of boundary-layer outflow to the WNAO. The plots show Hovmöller diagrams of GEOS-Chem daily mean air temperature (K), horizontal winds (m/s), CO (ppbv), and sea salt mixing ratios (ppbv) at 950 hPa along 72.5°W over the WNAO during Feb.-Mar. and Aug.-Sep. 2020. Arrows are wind vectors. Two horizontal dotted white lines indicate the latitude range (32.4-39.8°N) of aircraft measurements. Vertical white lines represent the days of cold front passages as visually identified by cold air intrusion from north of ~40°N.






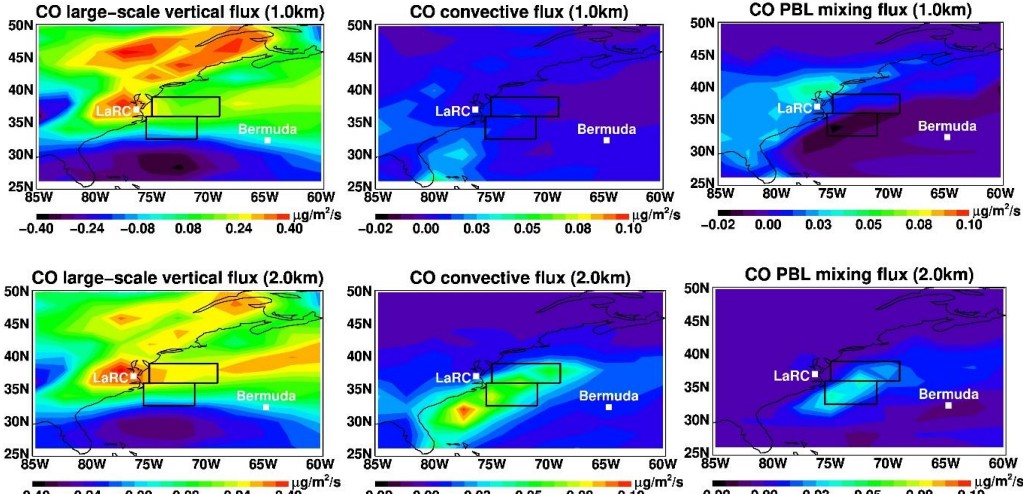

**Figure 4a.** Model simulated large-scale (resolved) vertical fluxes, convective fluxes, and PBL turbulent mixing fluxes of CO at 1
km (upper row) and 2 km (lower row), averaged over the period of Feb. 14 - Mar. 12, 2020. The two rectangular boxes denote
major flight areas (see "N" and "S" in Fig. 1).







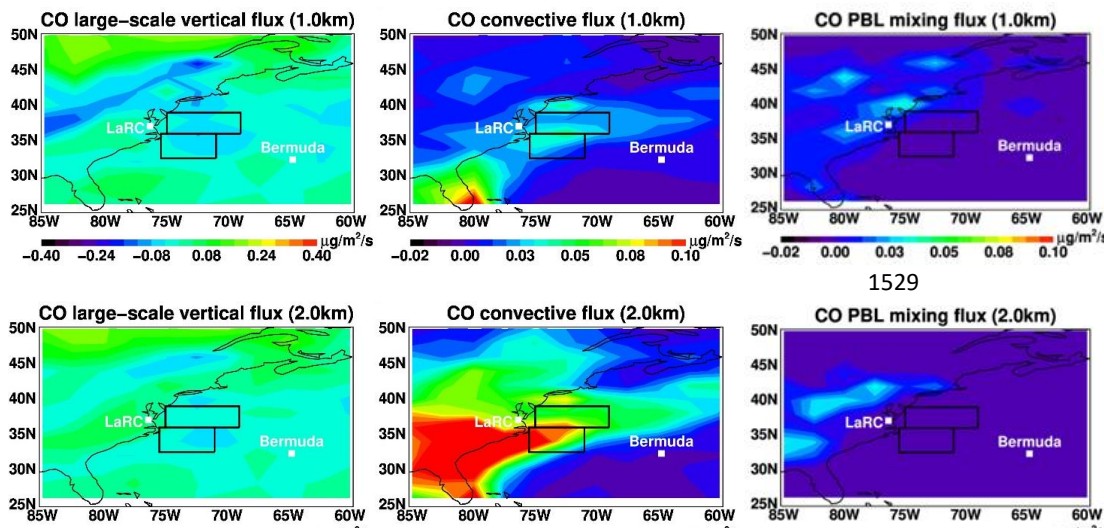



**Figure 4b**. Same as Fig. 4a, but for Aug. 13 - Sep. 30, 2020.








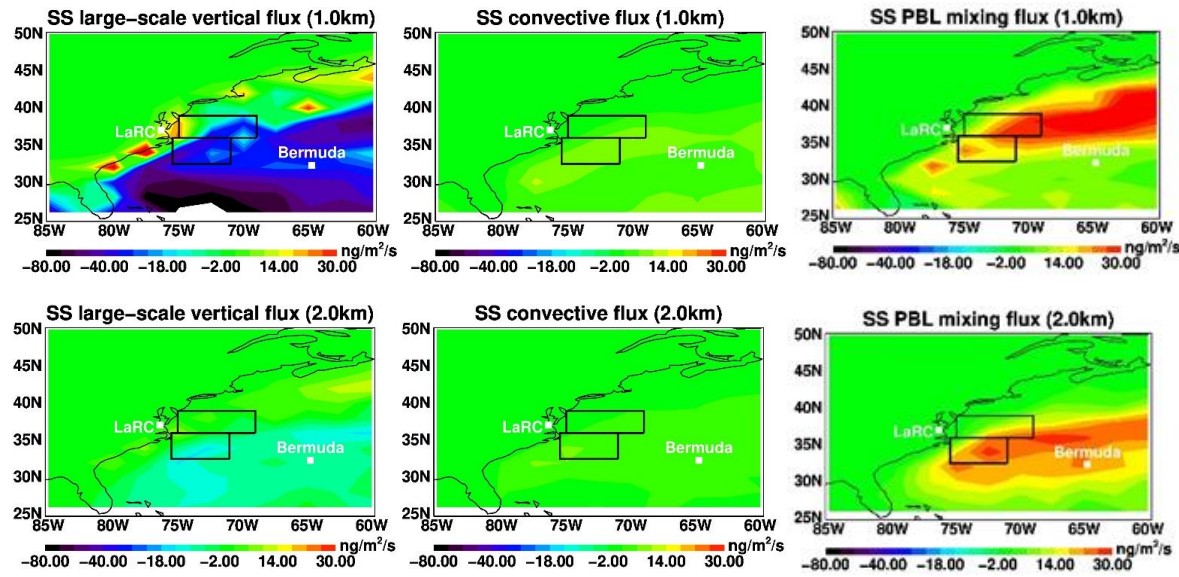


**Figure 4c**. Same as Fig. 4a, but for sea salt during Feb. 14 – Mar. 12, 2020.








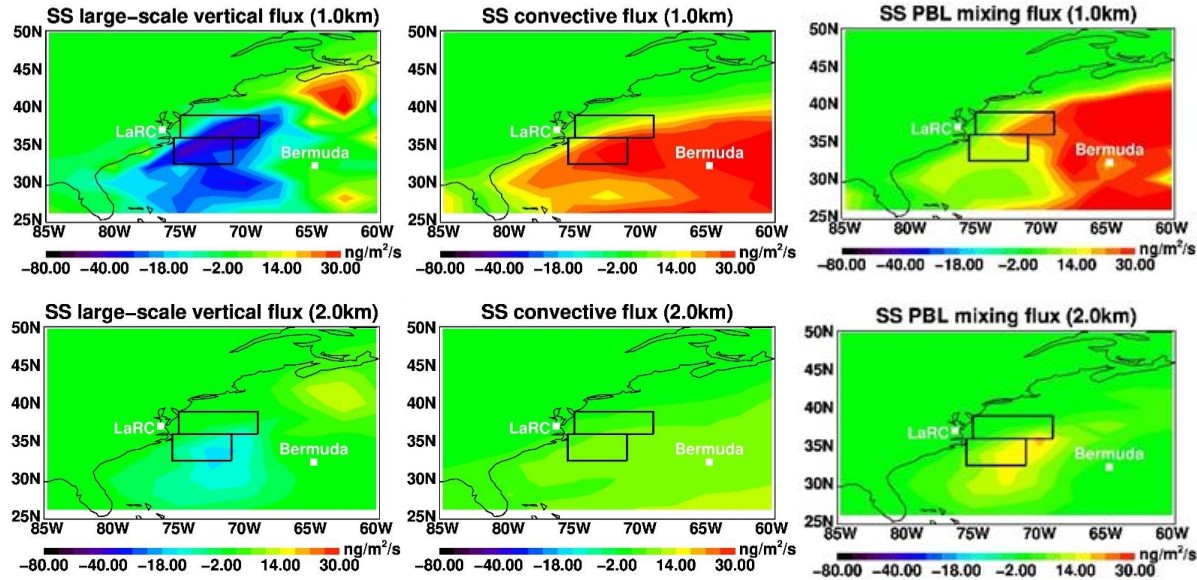


**Figure 4d**. Same as Fig. 4b, but for sea salt during Aug. 13 – Sep. 30, 2020.








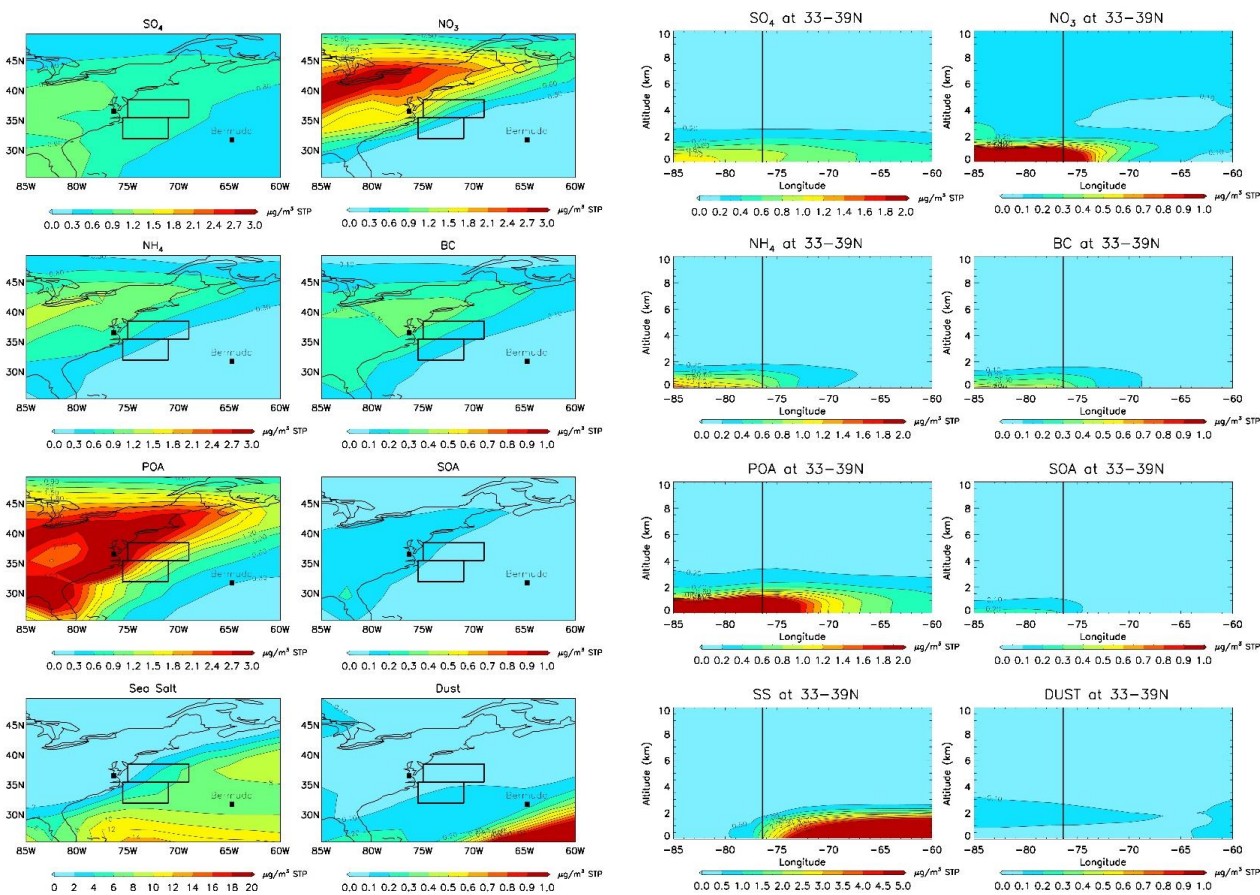


**Figure 5.** Model simulated concentrations of sulfate-nitrate-ammonium (SNA), black carbon (BC), primary organic aerosol (POA), secondary organic aerosol (SOA), sea salt, and dust averaged over the period of Feb. 14 - Mar. 12, 2020. Left two columns: map plots at 929 hPa; Right two columns: longitude-altitude cross-sections averaged over 33-39°N. Note different color scales among panels. Rectangles are the main areas sampled by aircraft during Feb.-Mar. and Aug.-Sep. 2020. The locations of LaRC and Bermuda are marked by black squares. The vertical lines in the right columns indicate the longitude (76.4°W) of LaRC.









**Figure 6.** Same as Figure 5, but for Aug. 13 - Sep. 30, 2020.


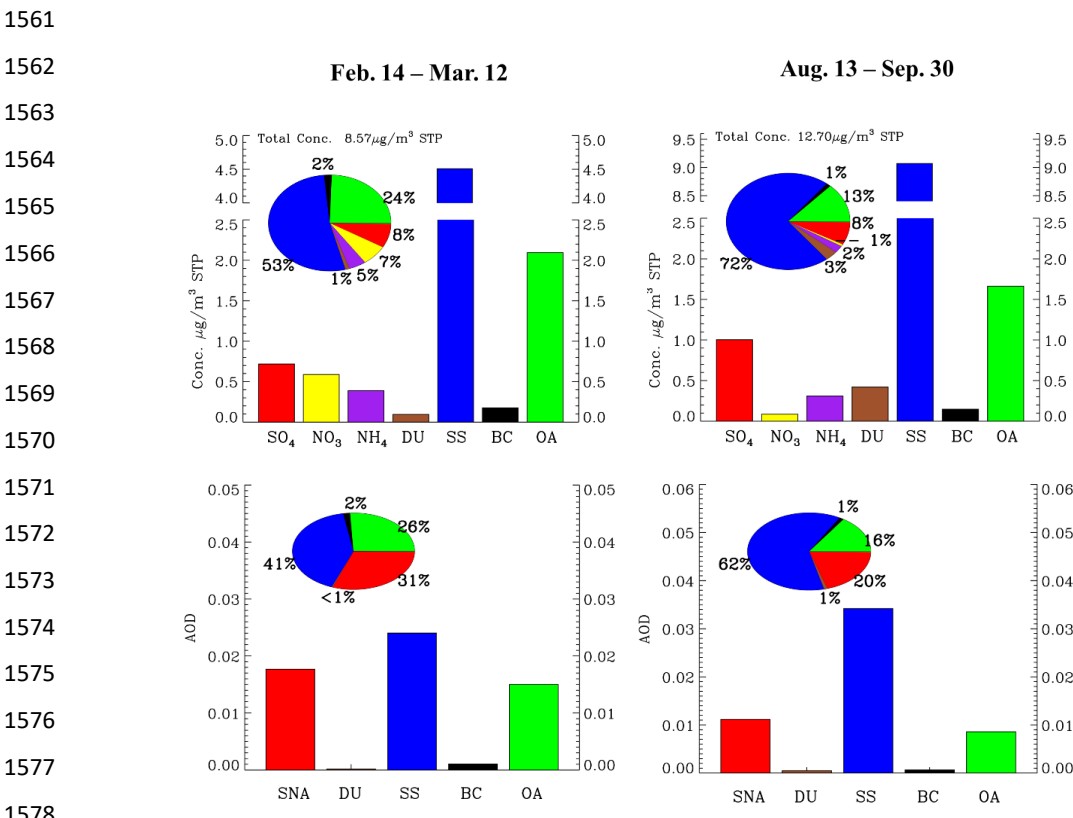

**Figure 7.** Model simulated mass concentrations (μg m⁻³ STP; upper row) and speciated (sulfate, nitrate, ammonium, dust, sea salt, black carbon, and organics) AOD in the boundary-layer (lower row) averaged over the flight areas (i.e., "N" and "S" in **Fig. 1**) during Feb. 14 - Mar. 12 (left column) and Aug. 13 - Sep. 30 (right column), 2020, respectively. Percentages on the pie chart denote the fractions of aerosol species to the total aerosol mass or AOD. Note that the model only calculates a combined AOD for sulfate, nitrate, and ammonium (SNA).





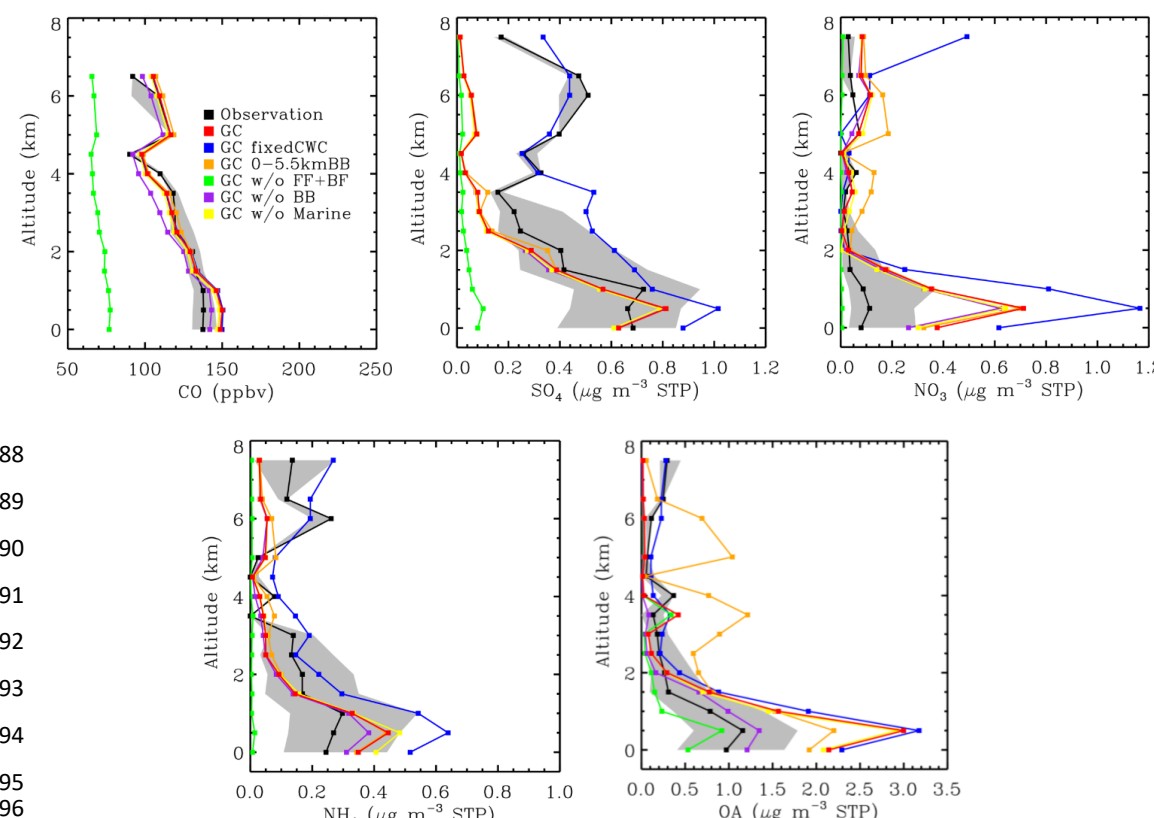

**Figure 8.** Comparison of model simulated (red) vertical profiles of CO (ppbv), sulfate, nitrate, ammonium, and organic aerosol
(OA; μg m$^{-3}$ STP) mixing ratios with Falcon aircraft measurements (black) during Feb.-Mar. 2020. Also shown are model results
from simulations (Table 1) with (1) a fixed value for cloud water content used in aerosol scavenging ("fixedCWC"), (2) biomass
burning emissions injected to the 0-5.5km altitudes, (3) fossil fuel and biofuel emissions turned off, (4) biomass burning emissions
turned off, or (5) marine emissions turned off, respectively. An OA/OC ratio of 2.1 (Philip et al., 2014) is used to convert simulated
OC to compare with AMS OA measurements. Hourly model output was sampled at the time and location of aircraft measurements.
Values (500m-binned) are medians over all flights. Gray shaded areas indicate the ranges of 25[th] – 75[th] percentiles for the
observations.






**Figure 9.** Same as **Fig. 8**, but for Aug.-Sep. 2020.



**Figure 10**. (1) Top and middle rows: Model simulated daily AOD (at 550nm) versus daily AOD measurements from three AERONET sites (NASA LaRC, GSFC, and Tudor Hill) during Feb.-Mar. (top) and Aug.-Sep. (middle) 2020, respectively. Shown for model results are the simulations (Table 1) "standard" (black line), "fixedCWC" (green line), "BB0-5.5km" (orange line), "noanthbf" (blue line), "nobb" (red lines), and "nomari" (cyan lines). (2) Bottom row: Scatterplots of model AODs from the "standard" simulation vs AERONET AODs at the three sites for Feb.-Mar. and Aug.-Sep. 2020, respectively. Red lines are linear regression lines. Grey lines are 1:1 line. Legends show regression line equations and Pearson correlation coefficients (R). For Aug.-Sep. 2020, six very large AERONET AOD values (0.8-1.1), for which the model failed to capture (<0.1), are excluded in the analysis.





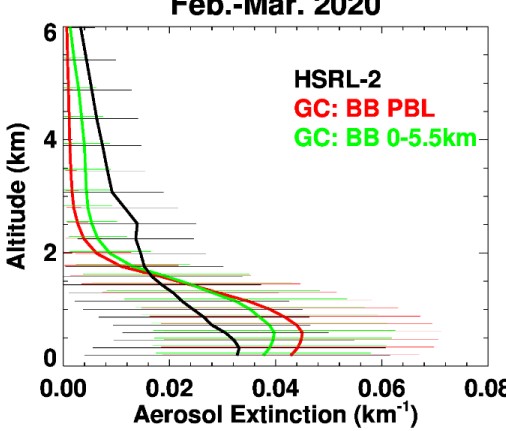
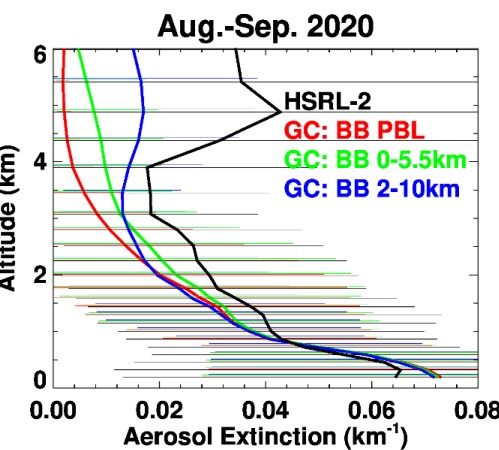

**Figure 11.** Comparisons of model aerosol extinctions (550nm) with aircraft HSRL-2 lidar measurements (532nm) averaged over
all flights during Feb.-Mar. and Aug.-Sep. 2020, respectively. Biomass burning emissions are injected into the planetary boundary
layer ("BB PBL"), into the 0-5.5 km altitude interval ("BB0-5.5km"), or into the 2-10 km altitude interval ("BB2-10km"). See
Table 1 for the configurations of model simulations. Hourly model output was sampled at the time and location of lidar
measurements. Horizontal lines denote +/- standard deviations of observed and simulated aerosol extinctions at model vertical
levels.


**Figure 12a**. Left column: Comparison of model results (550 nm) with vertical profiles of CALIOP aerosol extinction (532 nm)
averaged over the period of Feb. 14 – Mar. 12, 2020, and for four subdomains of the WNAO, respectively: North (39°-45°N, 72°-
62°W), Central (32°-39°N, 78°-68°W), and South (25°-32°N, 82°-72°W), as defined by Corral et al. (2020), and Bermuda (28°-
35°N, 69°-59°W). Note that three CALIOP data granules (2020-02-14T06-27-07ZN, 2020-03-03T06-21-54ZN, and 2020-03-
04T17-37-24ZD) are excluded from the North WNAO due to cloud contamination. Model results are sampled along the CALIPSO
orbit tracks in each box region. Right column: model speciated aerosol extinction profiles corresponding to the total aerosol
extinction profile on the left column. SNA = sulfate + nitrate + ammonium.







**Figure 12b**. Same as Fig. 12a, but for the period of Aug. 13 – Sep. 30, 2020. Also shown in the bottom two panels are the results for the North WNAO box region when biomass burning emissions are injected to the 0-5.5 km altitude interval ("BB0-5.5km", Table 1).





**Figure 13**. (a) Model simulated hourly total aerosol extinctions at ~1 km altitude over the WNAO during the King Air morning flight (14-17UTC) on March 12, 2020. Gray lines indicate the complete flight track with overlayed red lines denoting the flight tracks for each hour. (b) Time-height cross-section of aerosol extinctions observed by aircraft HSRL-2 lidar (532 nm) compared to that of model aerosol extinctions (550 nm) during the morning flight on Mar. 12, 2020. Curtain plots of model speciated aerosol extinctions along the flight track are also shown. SNA = sulfate + nitrate + ammonium.



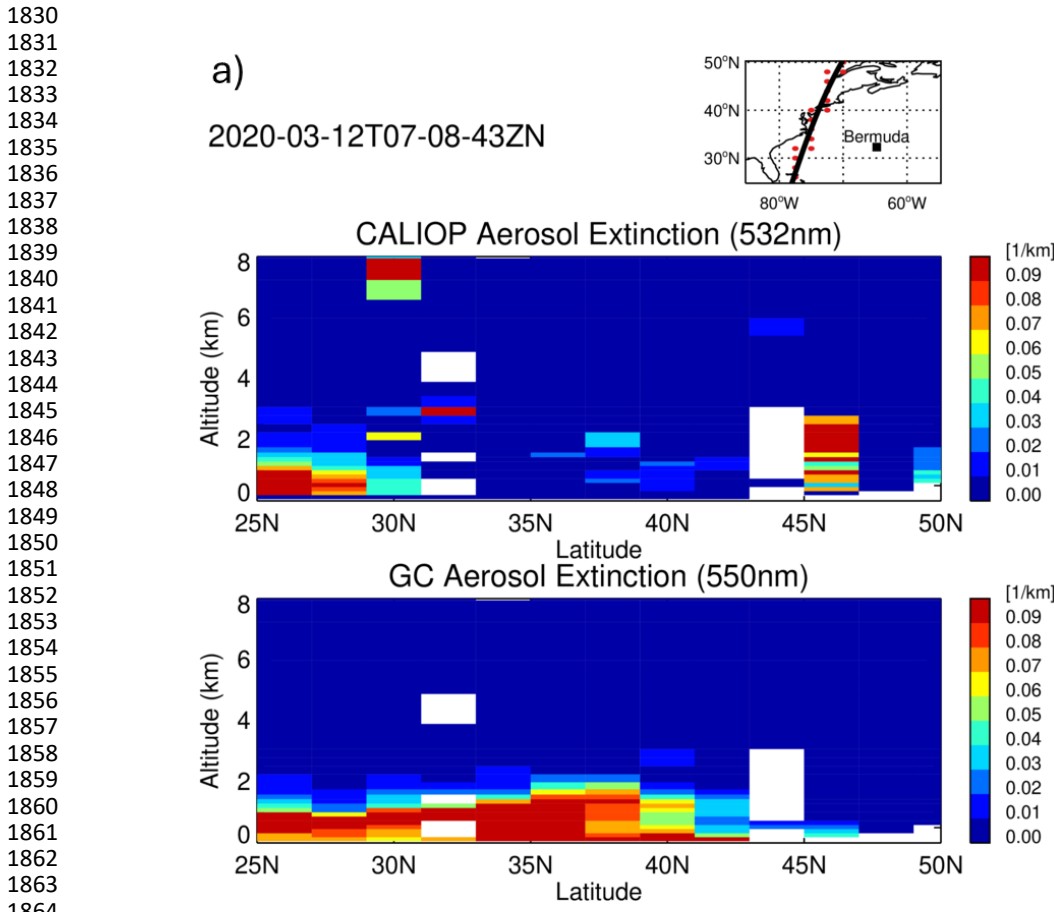

**Figure 14.** **(a)** Latitude-height cross-section of aerosol extinctions measured by CALIOP (532 nm) compared to that of model aerosol extinctions (550 nm) over the WNAO at 7:08UTC, Mar. 12, 2020. Model output is sampled at 1:30am LT. **(b)** Left column: Latitude-height cross-section of speciated aerosol extinctions (550 nm) over the WNAO along the CALIPSO orbit track at 7:08 UTC, Mar. 12, 2020, as simulated by the model. Model output is sampled at 1:30 am LT. Right column: same as left column, but for accumulated sea salt (SSa), coarse-mode sea salt (SSc), MERRA-2 RH(%), MERRA-2 cloud fraction, and MERRA-2 effective cloud extinction.




b)

**Figure 14.** (cont'd)



**Figure 15**. Same as Fig. 14ab but for the flight of Mar. 6, 2020.

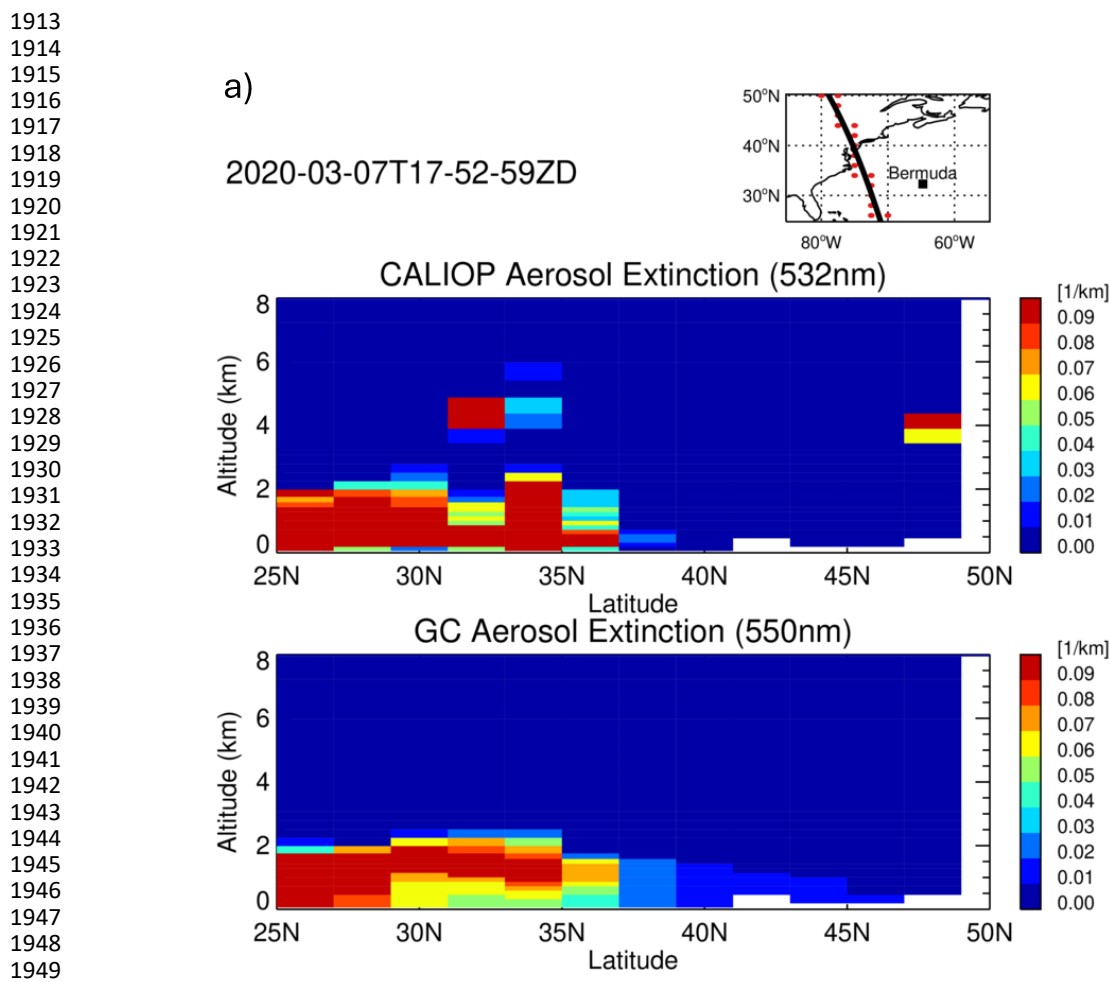

**Figure 16**. (a) Latitude-height cross-section of aerosol extinctions measured by CALIOP (532 nm) compared to that of model aerosol extinctions (550 nm) over the WNAO at 17:53 UTC, Mar. 7, 2020. Model output is sampled at 1:30 pm LT. (b) Left column: Latitude-height cross-section of speciated aerosol extinctions (550 nm) over the WNAO along the CALIPSO orbit track at 17:53 UTC, Mar. 7, 2020, as simulated by the model. Model output is sampled at 1:30 pm LT. Right column: same as left column, but for accumulated sea salt (SSa), coarse-mode sea salt (SSc), RH, cloud fraction, and effective cloud extinction.




b)



**Figure 16.** (cont'd)




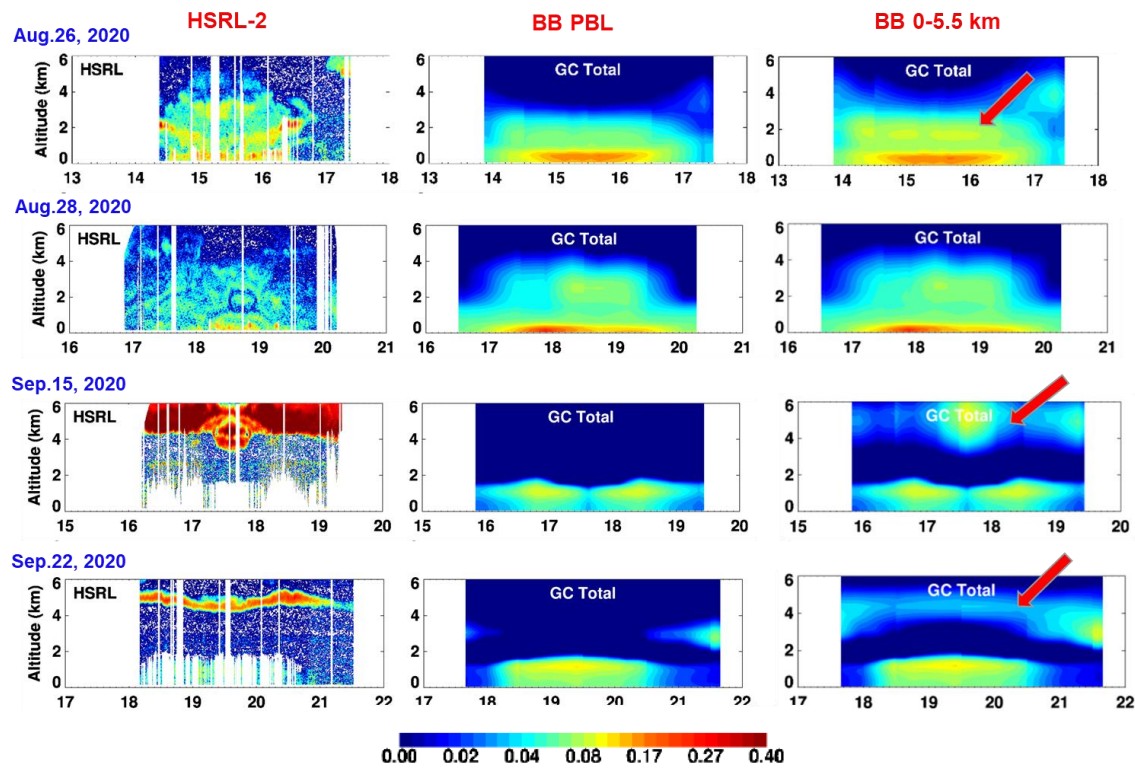

**Figure 17.** Selected cases where using 0-5.5 km fire emission injection heights improves the model simulations of HSRL-2 aerosol
extinction curtains during Aug.-Sep. 2020. The plots compare time-height cross-sections of aircraft HSRL-2 lidar aerosol
extinctions (532nm, left column) with those of model aerosol extinctions (550nm, middle and right columns) for the flights of Aug.
26, Aug. 28, Sep. 15, Sep. 22, 2020, respectively. Biomass burning emissions are injected into the planetary boundary layer ("BB
PBL", middle column) or into the 0-5.5km altitude interval ("BB 0-5.5km", right column). See Table 1 for details on model
simulations. Hourly model output was sampled at the time and location of lidar measurements.



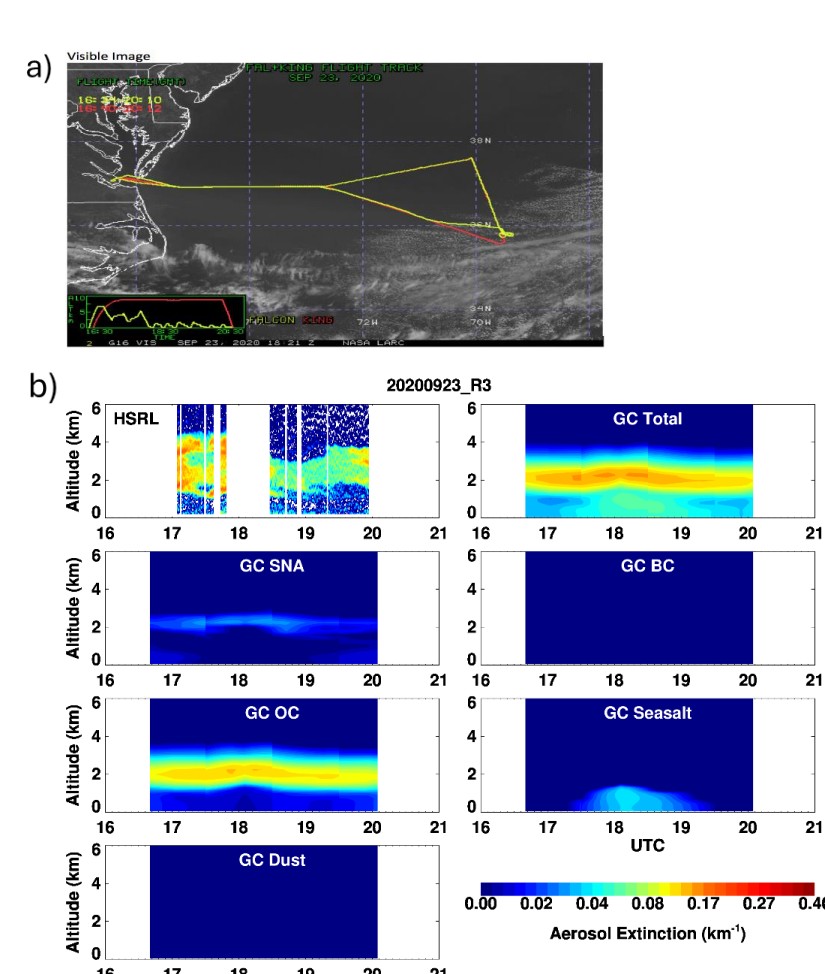

**Figure 18.** Case study for long-range transport of the western U.S. fire smoke to the WNAO on Sep. 23, 2020. (a) GOES-16 Quicklook Visible Images for 18:21 UTC, Sep. 23, 2020 (NASA Langley SATCORPS group). Superimposed are the King Air (red line) and Falcon (yellow line) flight tracks. (b) Time-height cross-section of aerosol extinctions observed by King Air HSRL-2 lidar (532 nm) compared to that of model aerosol extinctions (550 nm) for the flight of Sep. 23, 2020. There were no HSRL-2 measurements between 17:48-18:28UTC due to instrument issues. Also shown are curtain plots of model speciated aerosol extinctions along the flight track.

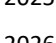

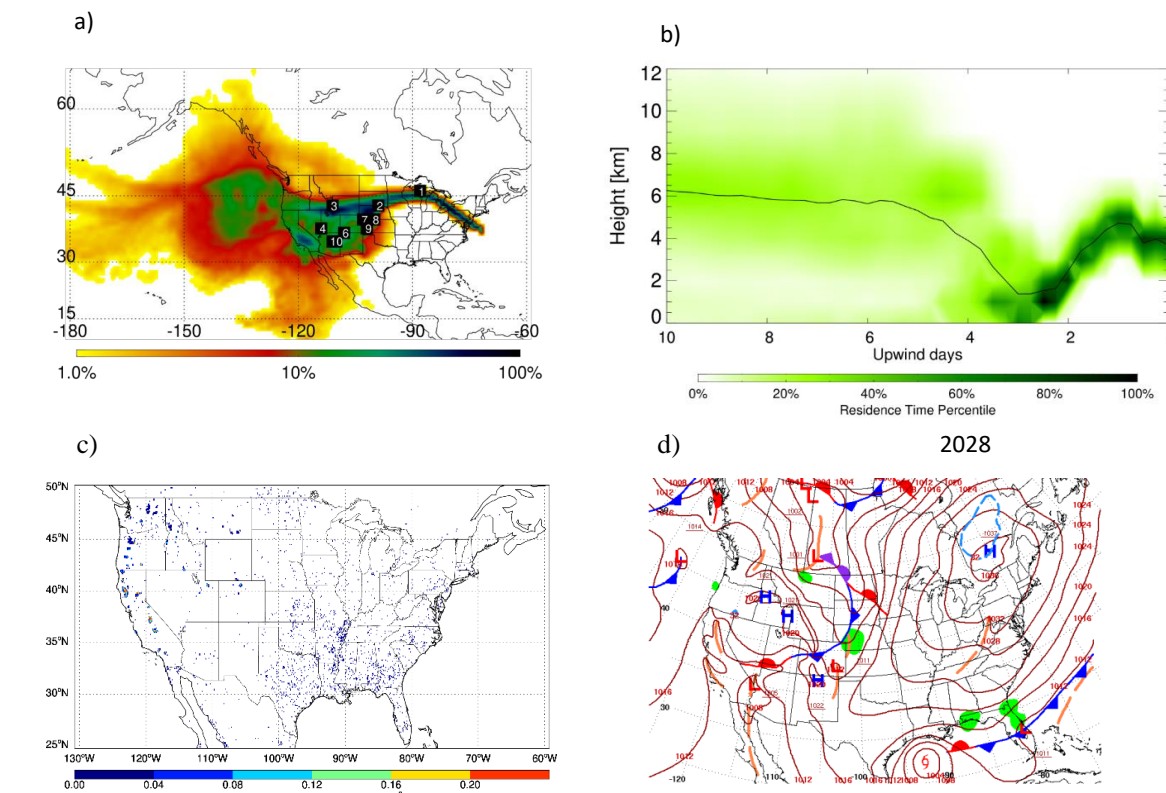

**Figure 19.** FLEXPART-simulated upwind air parcel residence time for the western U.S. fire smoke observed by HSRL-2 lidar over the WNAO at 17:13UTC, on Sep. 23, 2020. Panel (**a**) shows the column-integrated air parcel residence time during the entire simulated transport time (20 days). The white labels indicate the approximate locations of the center of the plumes on each upwind day up to the 10th day. Air parcels may pass the same location multiple times during transport and leave high density of residence time at those places. For better case-by-case comparison, residence time is color-coded by logarithmic grades representing its ratio to the location of maximal integrated residence time (100 %) during the transport. Panel (**b**) shows the vertical distribution of the residence time at given upwind times. The black line indicates the average height of air parcels during transport. (**c**) Monthly mean QFED biomass burning CO emissions for Sep. 2020. (**d**) Surface weather map at 7 am EST, Sep. 20, 2020 (https://www.wpc.ncep.noaa.gov/dailywxmap/index_20200920.html).





2041

2042

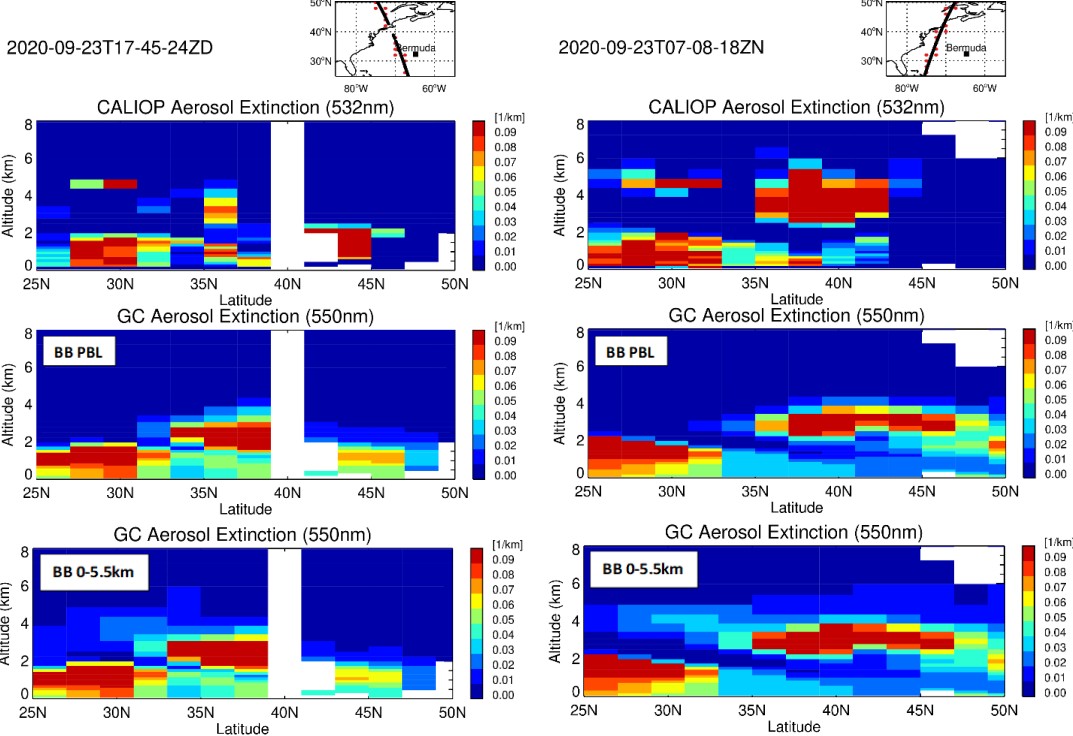

2043

**Figure 20**. Case study for the long-range transport of western U.S. fire smoke to the WNAO during Aug.-Sep. 2020. Latitude-height cross-section of aerosol extinctions measured by CALIOP (532 nm, top panel) compared to that of model aerosol extinctions (550 nm, middle and bottom panels) over the WNAO at 17:45 UTC (13:45 LT, left column) and 7:08 UTC (3:08 LT, right column), Sep. 23, 2020. Model output is sampled at 1:30 pm and 1:30 am LT, respectively. Biomass burning emissions are injected into the planetary boundary layer ("BB PBL", middle panel) or into the 0-5.5 km altitude interval ("BB 0-5.5km", bottom panel). See Table 1 for details on model simulations. The top right map shows the corresponding model grid-points (red dots) along the CALIPSO orbit track.

2051

2052





**Figure 21.** (a) Absolute changes (left column) and percentage changes (right column) in average AODs for the period of Feb.-Mar. 2020 when anthropogenic/biofuel emissions, biomass burning emissions, biogenic emissions, marine emissions, and dust emissions are respectively turned off in the model. The locations of LaRC and Bermuda are marked. (b) same as (a), but for the period of Aug-Sep. 2020. The two rectangular boxes denote major flight areas (see "N" and "S" in Fig. 1) of Feb.-Mar. and Aug.-Sep. 2020.





**Figure 21.** (cont'd)