# Peer review of "Tropospheric aerosols over the western North Atlantic Ocean during the winter and summer campaigns of ACTIVATE 2020"

_EGUsphere, 2024_

## Author Comment (AC1)

We thank the reviewers for their constructive comments, which helped improve the manuscript. Please find below our responses in blue font. Section and figure numbers refer to those in the revised manuscript.

**RC1: 'Comment on egusphere-2024-1127', Basudev Swain, 08 May 2024**

Review of "Tropospheric aerosols over the western North Atlantic Ocean during the winter and summer campaigns of ACTIVATE 2020: Life cycle, transport, and distribution" by Liu et al. (2024)

The manuscript provides valuable insights into the aerosol distributions and processes observed during the ACTIVATE 2020 campaign over the western North Atlantic Ocean. It effectively supports the campaign goal of studying aerosol-cloud-meteorology interactions by using data sets from all the different platforms, including GEOS-Chem model simulations, ground-based observations, and satellite data.

Moreover, the manuscript provides a comprehensive overview of the aerosol distribution, source, and transport coupled with meteorological parameters, along with an evaluation against ground-based and space-borne measurements. In addition, the authors perform sensitivity analyses with GEOS-Chem simulations, modifying parameters such as biomass burning height etc.

Overall, the manuscript demonstrates the satisfactory performance of the model against observations, especially in response to changing meteorological dynamics. It suggests potential improvements can be possible through finer resolution simulations.

I appreciate the authors for clearly defining the scientific objectives of the manuscript and for their valuable contributions to the understanding of aerosol dynamics, aerosol types, and associated meteorological drivers during the ACTIVATE campaign.

This manuscript falls well within the aim and scope of the ACP journal. I would recommend this manuscript for publication with minor corrections.

Total manuscript length is very high, authors need to work on reducing manuscript as well as abstract length. It would be helpful to future potential readers of this manuscript if the authors would consider these minor changes listed below:

Response – We thank Dr. Swain for the positive evaluation of the manuscript and the suggestions. We have shortened both the abstract and the text. Also see responses below for details.

General major questions:

1. The duration of the ACTIVATE campaign is six years (2019-2024), so why is this study focused only on the spring and summer of 2020?

Response – By focusing on the winter and summer deployments of 2020, we can compare the characteristics of tropospheric aerosols over the Western North Atlantic Ocean (WNAO) in the two seasons of the same year. Including more deployments in this analysis would make it difficult to present all results in one paper. We have added one sentence at the end of the first paragraph: "The model analysis of aerosols during the deployments of ACTIVATE 2021 and 2022 will be reported separately."

2. In this manuscript, it has been mentioned several times that the vertical transport can be improved by applying high-resolution models/simulations. Thus, in this manuscript the study domain is very small (two box regions: the North ("N"; 36-39°N, 69-75°W) and the South ("S"; 32.5-36°N, 71-75.5°W)), so why this study has not used finer resolution nested grid (0.25*0.3125) simulations provided by GEOS-Chem model instead of global simulations (2*2.5)? This finer resolution simulation could be used as another sensitivity simulation.

Response – Point is well taken. Nested grid simulations at finer resolution (e.g., with MERRA-2, $0.625°×0.5°$ longitude-by-latitude) would improve transport. Using GEOS-FP ($0.3125°×0.25°$ longitude-by-latitude) as input meteorology for both Feb.-Mar. 2020 and Aug.-Sep. 2020 is not preferrable since the FP system uses different convection parameterizations after June 1, 2020. Using global simulations at $2.5°×2°$ resolution allowed us to perform a series of perturbation experiments (Table 1) with computational efficiency. We plan to use the GEOS-Chem High Performance (GCHP) at $0.625°×0.5°$ resolution in a future modeling analysis of aerosol observations during ACTIVATE.

3. This study is conducted during August Complex "Gigafire" took place in mid-August 2020 and the California Creek fire occurred in early September 2020, ranked among the top five in California wildfire history. Thus to see the GEOS-Chem model comparisons during other years of ACTIVATE campaign would be very interesting, and could bring some valuable knowledge.

Response – Interesting point. The Aug.-Sep.2020 deployment was rescheduled from May-June 2020 due to the impact of COVID-19. The summer deployments of both 2021 and 2022 occurred during May-June prior to the peak season (late summer to fall) for California's wildfire activity. A follow-up paper is in preparation to report GEOS-Chem model analysis results for the May-June deployments of ACTIVATE 2021 and 2022. We have added the following sentences to the first paragraph: "The six field deployments took place during Feb.-Mar. and Aug.-Sep. 2020, Jan.-Apr. and May-Jun. 2021, Nov. 2021-Mar. 2022, and May-Jun. 2022, respectively." and "The model analysis of aerosols during the deployments of ACTIVATE 2021 and 2022 will be reported separately."

4. I was unable to find the method used in this manuscript to spatio-temporally collocate the GEOS-Chem model with ground-based, satellite data sets for evaluation. As models provide spatio-temporally continuous data, while ground, airborne, and satellite data are very discontinuous. So, what is the collocation strategy between all the datasets?

Response – As suggested below by the reviewer, a paragraph has been added to describe the method of model output sampling for each data set (subsection 2.3). This work does not attempt to collocate all the datasets used.

Minor comments:

Abstract:

The abstract is very long, it needs to be shortened further to make it easier to follow. Furthermore, there is a single big sentence from line 30 to line 34 that can be reduced to one small sentence, and mentioning GEOS-Chem model driven by MERRA-2 does not bring any additional information to the abstract, I suggest to remove MERRA-2 from the abstract.

Similarly, from line 43 to 46, this large sentence must be reduced in length.

Response – These specific suggestions are helpful. We have shortened the abstract accordingly. Lines 30-34 are revised to "We use the GEOS-Chem model to simulate aerosols that are evaluated against aircraft in situ and remote sensing measurements as well as ground-based and satellite observations over the WNAO during the winter (Feb. 14 – Mar. 12) and summer (Aug. 13 – Sep. 30) field deployments of ACTIVATE 2020." Lines 43-46 are revised to "Intensive aircraft measurements of non-refractory aerosols in the lower troposphere in winter provide useful constraints on model aerosol wet removal by precipitation scavenging".

Introduction:

1. Please cite some references at line 65 and 66.

Response – References have been added: "Aerosol particles scatter and absorb radiation in the atmosphere, directly or semi-directly affecting radiation budget and balance and thus climate (Charlson and Pilat, 1969; Hansen et al., 1997). Aerosols act as cloud condensation nuclei (CCN) or ice nuclei (IN), indirectly affecting radiation via the formation of clouds and precipitation (Twomey, 1974, 1977; Burrows et al., 2022). They also affect tropospheric photolysis and photochemistry by modifying solar radiation, and heterogeneous chemistry by providing surfaces for gas-particle interaction (Dickerson et al., 1997; Jacob, 2000; Martin et al., 2003). While the interaction of aerosols with clouds remains the largest uncertainty in the estimates of the Earth's changing energy budget, a full understanding requires knowledge of aerosol transport, sources, sinks, composition, and distribution, which still have large gaps (Boucher et al., 2013; Bellouin et al., 2019; Li et al., 2022)."

Bellouin, N., Quaas, J., Gryspeerdt,E., Kinne, S., Stier, P., Watson-Parris,D., et al. (2020). Bounding global aerosol radiative forcing of climate change. Reviews of Geophysics, 58, e2019RG000660. https://doi.org/10.1029/2019RG000660

Burrows, S. M., McCluskey, C. S., Cornwell, G., Steinke, I., Zhang, K., Zhao, B., et al. (2022). Ice-nucleating particles that impact clouds and climate: Observational and modeling research needs. Reviews of Geophysics, 60, e2021RG000745. https://doi.org/10.1029/2021RG000745

Charlson, R. J. and Pilat, M. J.: Climate: The influence of aerosols, J. Appl. Meteorol., 8, 1001–1002, 1969.

Dickerson, R. R., Kondragunta, S., Stenchikov, G., Civerolo, K. L., Doddridge, B. G., and Holben, B. N.: The impact of aerosols on solar ultraviolet radiation and photochemical smog, Science, 278, 827–830, https://doi.org/10.1126/science.278.5339.827, 1997.

Hansen, J. et al. Radiative forcing and climate response. Journal of Geophysical Research-Atmospheres **102**, 6831-6864 (1997).

Jacob, D.J., Heterogeneous chemistry and tropospheric ozone, Atmos. Environ., 34(2000), pp.2131-2159, 2000.

Li, J., Carlson, B.E., Yung, Y.L. et al. Scattering and absorbing aerosols in the climate system. Nat Rev Earth Environ **3**, 363–379 (2022). https://doi.org/10.1038/s43017-022-00296-7

Martin, R. V., D. J. Jacob, R. M. Yantosca, M. Chin, and P. Ginoux (2003), Global and regional decreases in tropospheric oxidants from photochemical effects of aerosols, J. Geophys. Res., 108, 4097, doi:10.1029/2002JD002622, D3.

Twomey, S. Pollution and the planetary albedo. Atmos. Environ. 8, 1251–1256 (1974).

Twomey, S. The influence of pollution on the shortwave albedo of clouds. J. Atmos. Sci. 34, 1149–1152 (1977).

2. The introduction is very large at about 4 pages. I would strongly suggest to reduce it. From line 79 to 115 has the potential to be reduced as there is no need for such large discussions NAO+, NAO-, and synoptic scale impact of cyclones on wind pattern and consequent impact on aerosol transport in the introduction. Just mention the direction of the wind pattern created by NAO oscillation and cyclones, and the associated aerosol transport in 3-4 lines.

Response –We have removed the discussions on NAO, which is not directly relevant to this study, and shortened the text on synoptic-scale transport.

4. Furthermore, all these NAO and cyclonic transports have been well presented with figures in Section 4 "Meteorological Settings and Transport Pathways", so why discuss them twice? Once in the Introduction and again in Section 4.

Response – See Response above. The relevant text has been significantly shortened to only summarize the current knowledge of transport pathways. The section of "Meteorological Settings and Transport Pathways" describes the conditions specific to the field campaign.

After the end of the introduction, the second section starts as GEOS-Chem Model. I would suggest to have a section like Data and Methods. In this section all data sets from different platforms can be presented as sub-sections. Further, at the end of the Data and Methods section, write a paragraph about the collocation strategy used for GEOS-Chem model evaluation with ground-based and space-based datasets etc.

Response – Good suggestion. Now section 2 ("Model and Data") has three subsections. Model simulations and output sampling are presented in subsection 2.3, which includes a paragraph on the collocation strategy: "Hourly, daily, and deployment-average model output are saved for analysis. For comparison with aircraft in situ and HSRL-2 lidar measurements, hourly model output is sampled at the time and location of the aircraft. For comparison with daily AOD measurements at the three AERONET sites, daily model output is sampled at the location of each site. When comparing with CALIPSO data, model output averaged over 1-2 h LT and 13-14 h LT is sampled at the date and location of nighttime and daytime CALIOP measurements (i.e., CALIPSO orbit tracks), respectively."

In Section 2 "Model Description", use only the information about the model simulations and the sensitivity simulations. Include the emission inventories used in an Appendix. This will help to reduce the length of the manuscript and will be easier for the readers.

Response – We have moved the descriptions of anthropogenic, biogenic, marine DMS, and lightning NOx emissions to the Supplement.

At Section 4 "Meteorological Settings and Transport Pathways", this section explains very well about the meteorological drivers for the aerosol transport and variability over WNAO region. So, please reduce the explanation in introduction about NAO+, NAO-, and synoptic impacts on aerosol distributions.

Response –We have removed the discussions on NAO, which is not directly relevant to this study, and shortened the text on synoptic-scale transport (also see Response to item 2 above).

Furthermore, I would like to suggest to include figures 2 to 4 in the appendix, as the meteorological variability is not the result of this manuscript, but rather an established fact that is captured by the MERRA-2 and GEOS-Chem model. This will further help to shape the manuscript. Also keep figure 14 in the supplement, and also I was unable to read the color bar values, please improve it.

Response – We decide to keep Fig. 2 and Fig. 3 in the main text since meteorological settings and boundary-layer outflow of pollution are both critical for understanding aerosol transport and distribution (section 4 in the revised manuscript). Most of Fig. 4 (vertical transport) has been moved to the Supplement. By "Figure 14", the reviewer may have meant Fig.4. The color bars for both Fig. 4 and Fig. 14 are improved.

Section 5 "Simulated Aerosols over the WNAO and Model Evaluations" contains the most important result of this manuscript. This section is very well written and easy to understand.

It would be very helpful to segregate the section 7 "Summary and conclusions" into two sections, as this manuscript brings very valuable conclusion that supports the ACTIVATE campaign and future modelling aspects related to this campaign. However, the summary overshadows the conclusion of this manuscript.

Overall, although the manuscript is very long and took me a few days to read, I enjoyed reading it and appreciate the scientific motivation behind this manuscript to support ACTIVATE campaign.

Response – We thank the reviewer for the positive comments. Instead of segregating the "Summary and conclusions" section into two sections, we have added three sub-headings "Results", "Implications", and "Future research" and shortened the summary component of the section.

**RC2: 'Comment on egusphere-2024-1127', Anonymous Referee #2, 10 May 2024**

This manuscript by Liu et al. compares GEOS-Chem model simulations against the observations conducted during the 2020 winter and summer campaigns of the Aerosol Cloud meTeorology Interactions oVer the western ATlantic Experiment (ACTIVATE) campaign. The study examined the aerosol life cycle, composition, transport pathways, and distribution in the western North Atlantic Ocean (WNAO).

The GEOS-Chem model results were compared against aircraft in situ and remote sensing measurements, ground-based observation and satellite observations. The study identified major transport mechanisms of aerosols over the WNAO; for example, the wintertime cold fronts can facilitate the continental outflow of aerosols, and boundary layer (BL) turbulent mixing can vertically transport sea salt aerosols. Based on the model simulation, biomass burning (BB) injection height was also found to influence the vertical distribution of aerosols.

The manuscript is a comprehensive overview of the research findings during ACTIVATE and highlights the importance of model simulations in understanding aerosol properties and transport. The manuscript fits the scope of Atmospheric Physics and Chemistry. However, the manuscript can be improved further by focusing on the region where the aircraft campaign was conducted and replotting some of the figures. I recommend the publication of the manuscript after the following revisions.

Response – We thank the reviewer for the positive evaluation and the suggestions which helped improve our manuscript.

General comments:

1. My major comment is that the manuscript can be more focused on comparing model results against aircraft and ground measurements, especially in the region where aircraft measurements were conducted (WNAO Central). The comparison between the model results and CALIOP observations was not well organized, and many subpanels of the figures were left undiscussed in

the manuscript. Moreover, the quality of the figures and discussions in the section associated with the CALIOP comparison is not as good as the earlier sections. The authors may consider moving the CALIOP results to another manuscript or discussing the CALIOP WNAO Central results only in this manuscript.

Response – Now we focus on the discussions on the CAIOP WNAO Central results. The figures are re-made to improve the quality. The CALIOP WNAO North/South/Bermuda figures are moved to the Supplement.

2. Overall, the figure quality in this manuscript is not high. It could be the issue of figure resolution, but the authors need to properly label the subpanels and place the figures side by side for comparison – especially comparing color maps. If the figures are not related to the main context of the manuscript, they need to be removed or relocated to the supplementary information. In several figures (e.g., Fig. 5 and Fig. 19), the letters cannot be read. The fonts of the figures are also not consistent. The color bar labels are overlapping (e.g., Figs. 14 and 16), the axis label does not have the proper unit (Figs. 14 and 16), or the unit is missing (Fig. 17). These issues need to be carefully addressed to maintain the scientific value of the manuscript.

Response – We thank the reviewer for these specific suggestions, each of which we have addressed. Conversion of figures from postscript to PDF degraded the quality to some extent but we will upload high-resolution images during the production process.

Detailed comments:

1. Line 123: The authors provided a detailed literature review for the sulfate-nitrate-ammonium (SNA), sea salt, and aerosols, but the discussion of OA and SOA is relatively short. The southeast U.S. is known for the large SOA yield due to the interaction between biogenic and anthropogenic emissions. The SOA from this region is expected to contribute to the WNAO due to the continental outflow.

Response – Excellent point. We have added the following text on SOA: "The southeast U.S. is known as a region for large biogenic SOA production amplified by anthropogenic emissions (Zheng et al., 2023; Zhang et al., 2018). The SOA from this region is expected to contribute to the continental outflow of aerosols to the WNAO (Ford and Heald, 2013)."

Zheng, Y., Horowitz, L. W., Menzel, R., Paynter, D. J., Naik, V., Li, J., and Mao, J.: Anthropogenic amplification of biogenic secondary organic aerosol production, Atmos. Chem. Phys., 23, 8993–9007, https://doi.org/10.5194/acp-23-8993-2023, 2023.

Zhang, H., L. D. Yee, et al., Monoterpenes are the largest source of summertime organic aerosol in the southeastern United States, PNAS, v.115, no.9, 2038-2043, 2018. http://www.pnas.org/cgi/doi/10.1073/pnas.1717513115

2. Line 174: The authors mentioned that there are "two campaigns." Is there another campaign apart from ACTIVATE? Or are the authors referring to "two deployments"?

Response – Changed "campaigns" to "deployments" throughout the text.

3. Line 313 to 336: I suggest moving this section to the supplementary information since it is not directly related to the discussions of the manuscript.

Response – We have moved this section to the Supplement, as suggested.

4. Figs. 2a and 2b: Please reorganize the figures for a side-by-side comparison between winter and summer measurements. It is difficult to compare color maps, especially when they are in different figures. Please improve the figure resolution because the letters and symbols cannot be clearly recognized (see general comment #2). Also, can the authors explain why the BL height is higher in winter?

Response – The figures are now organized for a side-by-side comparison between winter and summer measurements. The figure resolution is improved. We have added the following text in section 3 to explain why the BL height is higher in winter: "The generally higher marine BL height over the WNAO in winter (vs summer) is mainly due to a larger temperature contrast between the relatively warm ocean surface and the colder air above, leading to increased lower-atmospheric instability and turbulent mixing and thus a higher BL height (Chien et al., 2019; Gallo et al., 2023; Wang et al., 2021)."

Chien, F.-C., J.-S. Hong, and Y.-H. Kuo, The marine boundary layer height over the Western North pacific based on GPS radio occultation, island soundings, and numerical models, Sensors, 19, 155; doi:10.3390/s19010155, 2019.

Gallo, F., Sanchez, K. J., Anderson, B. E., Bennett, R., Brown, M. D., Crosbie, E. C., Hostetler, C., Jordan, C., Yang Martin, M., Robinson, C. E., Russell, L. M., Shingler, T. J., Shook, M. A., Thornhill, K. L., Wiggins, E. B., Winstead, E. L., Wisthaler, A., Ziemba, L. D., and Moore, R. H.: Measurement report: Aerosol vertical profiles over the western North Atlantic Ocean during the North Atlantic Aerosols and Marine Ecosystems Study (NAAMES), Atmos. Chem. Phys., 23, 1465–1490, https://doi.org/10.5194/acp-23-1465-2023, 2023.

Wang, J., Wood, R., Jensen, M. P., Chiu, J. C., Liu, Y., Lamer, K., Desai, N., Giangrande, S. E., Knopf, D. A., Kollias, P., Laskin, A., Liu, X., Lu, C., Mechem, D., Mei, F., Starzec, M., Tomlinson, J., Wang, Y., Yum, S. S., Zheng, G., Aiken, A. C., Azevedo, E. B., Blanchard, Y.,

China, S., Dong, X., Gallo, F., Gao, S., Ghate, V. P., Glienke, S., Goldberger, L., Hardin, J. C., Kuang, C., Luke, E. P., Matthews, A. A., Miller, M. A., Moffet, R., Pekour, M., Schmid, B., Sedlacek, A. J., Shaw, R. A., Shilling, J. E., Sullivan, A., Suski, K., Veghte, D. P., Weber, R., Wyant, M., Yeom, J., Zawadowicz, M., and Zhang, Z.: Aerosol and Cloud Experiments in the Eastern North Atlantic (ACE-ENA), B. Am. Meteorol. Soc., 1–51, https://doi.org/10.1175/BAMS-D-19-0220.1, 2021.

5. Line 399: The authors discussed different transport mechanisms, both in the horizontal and vertical directions. However, in the horizontal direction, are there other mechanisms other than cold fronts? In the vertical direction, it is not clear how uplifting ahead of cold fronts and convective transport are different from each other. In addition, is entrainment considered in the vertical transport mechanisms? It is an efficient mechanism to control the aerosol population in the Eastern North Atlantic (ENA, see reference below), which is also affected by the continental outflow from North America.
Zheng, G., Wang, Y., Aiken, A. C., Gallo, F., Jensen, M. P., Kollias, P., ... & Wang, J. (2018). Marine boundary layer aerosol in the eastern North Atlantic: seasonal variations and key controlling processes. Atmospheric Chemistry and Physics, 18(23), 17615-17635.

Response – Besides the frontal passages, prevalent westerly flows are also a mechanism to transport pollution from the North American continent to WNAO. However, westerly flows alone have a limited impact on the WNAO unless they are accompanied by other weather systems (e.g., warm conveyor belt associated with midlatitude cyclones). We have added the following text: "… resulting in enhanced CO during September 23-25. In addition, prevalent westerly flows, particularly during winter and spring, are a significant mechanism for transporting pollution from the North American continent to the WNAO (Sorooshian et al., 2020). However, effective transport of these pollutants to the WNAO typically requires the westerly flow to be coupled with midlatitude cyclones. These cyclones facilitate the outflow of pollution in the BL and the lifting of BL air from the North American continent into the free troposphere over the WNAO (Creilson et al., 2003)."

While the large-scale uplifting ahead of cold fronts is resolved by the model, the subgrid (unresolved) convection transport requires being parameterized. We state in section 3: "We analyzed model resolved large-scale vertical fluxes, convective fluxes, and PBL turbulent mixing fluxes …".

Entrainment is included in the PBL turbulent mixing process in the model, which is considered a vertical transport mechanism in this study. We have added a few sentences to discuss free-tropospheric entrainment: "…to the free troposphere in summer (Fig. 4; Fig. S1c). Entrainment (i.e., turbulent mixing of air from the free troposphere into the BL) was previously shown to be the major source of the MBL aerosol population in the Eastern North Atlantic (Zheng et al.,

2018). Tornow et al. (2022) recently emphasized the important role of entraining free-tropospheric clean air in diluting MBL CCN under cold air outbreak conditions during the ACTIVATE field campaign. Our model-calculated PBL mixing flux of CO in the upper BL (~ 1.0 km) are negative (downward) in the S box region, suggesting entrainment plays a role there (Fig. S1a)."

Creilson, J. K., Fishman, J., & Wozniak, A. E. (2003). Intercontinental transport of tropospheric ozone: a study of its seasonal variability across the North Atlantic utilizing tropospheric ozone residuals and its relationship to the North Atlantic Oscillation. Atmospheric Chemistry and Physics, 3(6), 2053-2066.

Sorooshian, A., Corral, A. F., Braun, R. A., Cairns, B., Crosbie, E., Ferrare, R., ... & Zuidema, P. (2020). Atmospheric research over the Western North Atlantic Ocean Region and North American East Coast: A review of past work and challenges ahead. Journal of Geophysical Research: Atmospheres, 125(6), e2019JD031626.

Tornow, F., Ackerman, A. S., Fridlind, A. M., Cairns, B., Crosbie, E. C., Kirschler, S., et al. (2022). Dilution of boundary layer cloud condensation nucleus concentrations by free tropospheric entrainment during marine cold air outbreaks. Geophysical Research Letters, 49, e2022GL098444. https://doi.org/10.1029/2022GL098444

Zheng, G., Wang, Y., Aiken, A. C., Gallo, F., Jensen, M. P., Kollias, P., ... & Wang, J. (2018). Marine boundary layer aerosol in the eastern North Atlantic: seasonal variations and key controlling processes. Atmospheric Chemistry and Physics, 18(23), 17615-17635.

6. Fig. 4c: The first panel (please label each panel inside the figure) shows that there is a high large-scale downward vertical flux for sea salt aerosols. Is this result true? What does it imply?

Response –We thank the reviewer for pointing out this error. The diagnostic for the large-scale vertical fluxes in this version of the standard model code had issues and was inadvertently not updated for this study. This is now corrected (Fig. 4 and Fig. S1c).

7. Line 423: Why is biofuel considered in the model? Is it a major source of aerosols? Also, should it be considered as a part of anthropogenic source?

Response – Biofuel emissions are a significant source of aerosols and are separately treated in the GEOS-Chem model. They are now included as part of the anthropogenic sources in the text.

8. Line 450: The model simulation shows that the sea salt aerosol mass concentration is higher in summer. Is it driven by the higher wind speed in the summer? Can the authors provide the average wind speeds during the deployments?

Response – Good suggestion. As discussed in section 3, "Two strong events of high sea salt concentrations occurred on September 10-12 and 19-22. The former was due to high surface winds associated with a westward moving low-pressure system. The latter resulted from the strong winds during a cold air outbreak that lasted for several days when cold air swept along a NE-to-SW corridor off the east coast over the whole flight domain. It lasted until a new cold front moved into the domain on September 23, resulting in enhanced CO during September 23-25." Now we have revised the text in the last paragraph of section 4.1 to the following: "Sea salt is a dominant fraction of total aerosol mass in the BL in both winter (53%) and summer (72%), followed by organics with 24% in winter and 13% in summer. The higher sea salt mass concentration is also the main reason for higher total aerosol mass in summer. The former is ascribed to two events of strong surface winds and high sea salt emissions during September 10-12 and 19-22 (section 3), despite the lower average surface wind speed over the domain of (32-40°N, 62-76°W) during the summer (1.2 m/s) vs winter (3.7 m/s) deployment."

9. Figs. 5 and 6: The letters cannot be recognized in the figure. Please improve the figure quality.

Response – Revised accordingly.

10. Fig. 8 (last panel): Can the authors discuss why there are two peaks in OA under 0 to 5.5 km BB condition?

Response –We have added the following sentence in the text: "The occurrence of two peaks is likely due to the bifurcation of biomass burning plumes associated with transport in each layer."

11. Line 491: Why would a smaller CWC lead to a faster conversion of cloud water to precipitation? Should it be the opposite?

Response –We have changed the text to read: "Conversion of cloud water to precipitation is determined by the ratio of rainwater / CWC. Since rainwater uses the same value from MERRA-2, the smaller CWC from MERRA-2 compared to the assumed fixed value ($1.0 \times 10^{-3}$ kg m$^{-3}$, Table 1) results in a faster conversion from cloud water to precipitation."

12. Line 511: The authors stated that the reason why the model cannot reproduce the observed CO concentration is because of the model's inability to retain the western U.S. BB plumes. However, the model agrees reasonably well with the observed BB aerosols, especially when

using different injection heights. Are there additional reasons for the disagreement in the CO vertical profile? The authors may consider checking CO and BC measurements over the ENA, which is also under the influence of North America continental outflows (references below).

Parrish, D. D., Trainer, M., Holloway, J. S., Yee, J. E., Warshawsky, M. S., Fehsenfeld, F. C., ... & Moody, J. L. (1998). Relationships between ozone and carbon monoxide at surface sites in the North Atlantic region. Journal of Geophysical Research: Atmospheres, 103(D11), 13357-13376. Wang, Y., Zheng, G., Jensen, M. P., Knopf, D. A., Laskin, A., Matthews, A. A., ... & Wang, J. (2021). Vertical profiles of trace gas and aerosol properties over the eastern North Atlantic: variations with season and synoptic condition. Atmospheric Chemistry and Physics, 21(14), 11079-11098.

Response – We appreciate the reviewer's insight. The following text has been added:

"… CO panel of Fig. 9). The primary reason for the underestimated CO in GEOS-Chem is likely the model's excessive OH concentrations, which accelerate CO oxidation and reduce overall CO mixing ratios. Jin et al. (2023) also highlighted that GEOS-Chem tends to underestimate concentrations of highly reactive VOCs, which likely contributes to the OH overestimate. In a comparison with airborne observations of wildfire emissions, Carter et al. (2021) found that while the model simulates well aerosol concentrations, the low CO bias suggests potential issues related to the representation of chemical processes in the model. In addition, uncoupled $CH_4$, CO, and $CO_2$ chemistry could lead to the OH biases in GEOS-Chem (Bukosa et al., 2023). Nevertheless, clear reductions of CO in the model simulations …"

We would also like to note that our simulations with different injection heights were exploratory tests designed to demonstrate that higher injection heights for large fires could improve simulated CO. The improved OA results for BB plumes (Fig. 9) should be considered qualitative rather than quantitative.

Regarding the studies by Parrish et al. (1998) and Wang et al. (2021), while they focus on observational data and provide valuable insights into atmospheric processes over the ENA region, their relevance to our study is limited. ENA is more remote compared to WNAO. CO simulations over the ENA may be influenced by different modeling uncertainties. Since neither study discusses CO modeling directly, we have decided not to include discussions about these two studies.

Carter, T. S., Heald, C. L., Cappa, C.D., Kroll, J. H., Campos, T. L., Coe, H.,et al. (2021). Investigating carbonaceous aerosol and its absorption properties from fires in the western United States (WE-CAN) and southern Africa (ORACLES and CLARIFY). Journal of Geophysical Research: Atmospheres,126, e2021JD034984. https://doi.org/10.1029/2021JD034984

Jin, L., Permar, W., Selimovic, V., Ketcherside, D., Yokelson, R. J., Hornbrook, R. S., Apel, E. C., Ku, I.-T., Collett Jr., J. L., Sullivan, A. P., Jaffe, D. A., Pierce, J. R., Fried, A., Coggon, M. M., Gkatzelis, G. I., Warneke, C., Fischer, E. V., and Hu, L.: Constraining emissions of volatile organic compounds from western US wildfires with WE-CAN and FIREX-AQ airborne observations, Atmos. Chem. Phys., 23, 5969–5991, https://doi.org/10.5194/acp-23-5969-2023, 2023.

Bukosa, B., Fisher, J. A., Deutscher, N. M., & Jones, D. B. (2023). A Coupled CH4, CO and CO2 Simulation for Improved Chemical Source Modeling. Atmosphere, 14(5), 764.

13. Line 531: The authors mentioned "weak entrainment" here. Is entrainment considered a vertical transport mechanism?

Response – As mentioned above, entrainment (i.e., turbulent mixing of air from the free troposphere into the boundary layer) is included in the PBL turbulent mixing process in the model, which is considered a vertical transport mechanism in this study.

14. Line 640: The authors can just use "Fig. 12" instead of "Fig. 12ab" because there are only two panels in the figure. Also, the left panels WNAO South and WNAO Bermuda in Fig. 12a are not discussed in the manuscript (please see general comment #2).

Response – "Fig. 12" is now used. We have added discussions on the panels for WNAO South/Bermuda (panels moved to the Supplement).

15. Fig. 13: Please consider reorganizing the figures. The flight tracks and flight directions cannot be clearly seen. If GC BC and Dust are negligible, these two panels can be removed from the main body of the manuscript and simply discussed in the manuscript.

Response – Good suggestions. We have re-plotted Fig. 13 with respect to flight tracks and directions. The model BC and dust panels have been removed from the plot.

16. Line 672: If the CALIOP retrieval results are in doubt, the discussion and figure should not be included in the manuscript since we cannot confidently compare the model and observations.

Response –Point is well taken. We have moved the CALIOP Figure 14 to the Supplement and mentioned it briefly in the text.

17. Line 679: Please just use "Fig. 15" instead of "Fig. 15ab."

Response –Revised accordingly.

18. Line 683: Please see the comment on Line 672.

Response – HSRL-2 provided meaningful retrievals for much of this flight. We have changed the statement to read "The HSRL-2 measurements of the general pattern of BL aerosol extinction over land and ocean is very similar to the model result (Fig. 13b)."

19. Section 5.4: This section can be restructured by having a subsection to (1) 5.4.1: show HSRL-2 and CALIOP results and (2) 5.4.2: conduct case analysis of land-ocean aerosol extinction gradient, SNA and sea salt mixing, transport of fire smoke, and so on. Please use subsections to divide the contents.

Response – Good suggestion. Now we use subsections 4.4.1 and 4.4.2 (in the revised manuscript) to divide the contents.

20. Line 739: Since the transport event happened over several days, would the one-time surface weather map be able to support the transport trajectories? The FLEXPART results should be sufficient to support long-range transport.

Response –The surface weather map has been removed, as suggested.